# Distribution Matching for Crowd Counting

**Boyu Wang**$^*$    **Huidong Liu**$^*$    **Dimitris Samaras**    **Minh Hoai**
Department of Computer Science, Stony Brook University, Stony Brook, NY 11790
{boywang, huidliu, samaras, minhhoai}@cs.stonybrook.edu
$^*$indicates equal contribution

## Abstract

In crowd counting, each training image contains multiple people, where each person is annotated by a dot. Existing crowd counting methods need to use a Gaussian to smooth each annotated dot or to estimate the likelihood of every pixel given the annotated point. In this paper, we show that imposing Gaussians to annotations hurts generalization performance. Instead, we propose to use Distribution Matching for crowd COUNTing (DM-Count). In DM-Count, we use Optimal Transport (OT) to measure the similarity between the normalized predicted density map and the normalized ground truth density map. To stabilize OT computation, we include a Total Variation loss in our model. We show that the generalization error bound of DM-Count is tighter than that of the Gaussian smoothed methods. In terms of Mean Absolute Error, DM-Count outperforms the previous state-of-the-art methods by a large margin on two large-scale counting datasets, UCF-QNRF and NWPU, and achieves the state-of-the-art results on the ShanghaiTech and UCF-CC50 datasets. DM-Count reduced the error of the state-of-the-art published result by approximately 16%.
Code is available at `https://github.com/cvlab-stonybrook/DM-Count`.

## 1  Introduction

Image-based crowd counting is an important research problem with various applications in many domains including journalism and surveillance. Current state-of-the-art methods [54, 8, 25, 55, 61, 59, 17, 48, 21, 23, 36] treat crowd counting as a density map estimation problem, where a deep neural network first produces a 2D crowd density map for a given input image and subsequently estimates the total size of the crowd by summing the density values across all spatial locations of the density map. For images of large crowds, this density map estimation approach has been shown to be more robust than the detection-then-counting approach [22, 19, 62, 12] because the former is less sensitive to occlusion and it does not need to commit to binarized decisions at an early stage.

A crucial step in the development of a density map estimation method is the training of a deep neural network that maps from an input image to the corresponding annotated density map. In all existing crowd counting datasets [15, 60, 14, 51], the annotated density map for each training image is a sparse binary mask, where each individual person is marked with a single dot on their head or forehead. The spatial extent of each person is not provided, due to the laborious effort needed for delineating the spatial extent, especially when there is too much occlusion ambiguity. Given training images with dot annotation, training the density map estimation network is equivalent to optimizing the parameters of the network to minimize a differentiable loss function that measures the discrepancy between the predicted density map and the dot-annotation map. Notably, the former is a dense real-value matrix, while the later is a sparse binary matrix. Given the sparsity of the dots, a function that is defined based on the pixel-wise difference between the annotated and predicted density maps is hard to train because the reconstruction loss is heavily unbalanced between the 0s and 1s in the sparse binary matrix. One approach to alleviate this problem is to turn each annotated dot into a Gaussian blob such that the ground truth is more balanced and thus the network is easier to train. Almost all prior crowd density map estimation methods [56, 57, 60, 38, 20, 33, 35, 4, 28, 50, 27, 40, 29, 26] have followed

this convention. Unfortunately, the performance of the resulting network is highly dependent on the quality of this "pseudo ground truth", but it is not trivial to set the right widths for the Gaussian blobs given huge variation in the sizes and shapes of people in a perspective image of a crowded scene.

Recently, Ma et al. [31] proposed a Bayesian loss to measure the discrepancy between the predicted and the annotated density maps. This method transforms a binary ground truth annotation map into $N$ "smoothed ground truth" density maps, where $N$ is the count number. Each pixel value of a smoothed ground truth density map is the posterior probability of the corresponding annotation dot given the location of that pixel. Empirically, this method has been shown to outperform other aforementioned approaches [60, 38, 20, 33, 35, 4]. However, there are two major problems with this loss function. First, it also requires a Gaussian kernel to construct the likelihood function for each annotated dot, which involves setting the kernel width. Second, this loss corresponds to an underdetermined system of equations with infinitely many solutions. The loss can be 0 for many density maps that are not similar to the ground truth density map. As a consequence, using this loss for training can lead to a predicted density map that is very different from the ground truth density map.

In this paper, we address the shortcomings in existing approaches with the following contributions.

- We theoretically and empirically show that imposing Gaussians to annotations will hurt the generalization performance of a crowd counting network.
- We propose DM-Count, a method that performs Distribution Matching for crowd COUNTing. Unlike previous works, DM-Count does not need any Gaussian smoothing ground truth annotations. Instead, we use Optimal Transport (OT) to measure the similarity between the normalized predicted density map and the normalized ground truth density map. To stabilize the OT computation, we further add a Total Variation (TV) loss.
- We present the generalization error bounds for the counting loss, OT loss, TV loss and the overall loss in our method. All the bounds are tighter than those of the Gaussian smoothed methods.
- Empirically, our method improved the state-of-the-art by a large margin on four challenging crowd counting datasets: UCF-QNRF, NWPU, ShanghaiTech, and UCF-CC50. Notably, our method reduced the published state-of-the-art MAE on the NWPU dataset by approximately 16%.

## 2 Previous Work

### 2.1 Crowd Counting Methods

Crowd counting methods can be divided into three categories: detection-then-count, direct count regression, and density map estimation. Early methods [22, 19, 62, 12] detect people, heads, or upper bodies in the image. However, accurate detection is difficult for dense crowds. Besides, it also requires bounding box annotation, which is a laborious and ambiguous process due to heavy occlusion. Later methods [5, 6, 49, 7] avoid the detection problem and directly learn to regress the count from a feature vector. But their results are less interpretable and the dot annotation maps are underutilized. Most recent works [20, 35, 15, 4, 31, 50, 27, 40, 29, 26, 54, 8, 25, 55, 61, 59, 17, 48, 21, 23, 30, 47, 53, 43, 37, 56, 18, 39, 24, 44, 42] are based on density map estimation, which has been shown to be more robust than detection-then-count and count regression approaches.

Density map estimation methods usually define the training loss based on the pixel-wise difference between the Gaussian smoothed density map and the predicted density map. Instead of using a single kernel width to smooth the dot annotation, [60, 14, 47] used adaptive kernel width. The kernel width is selected based on the distance to an annotated dot's nearest neighbors. Specifically, [15] generated multiple smoothed ground truth density maps on different density levels. The final loss combines the reconstruction errors from multiple density levels. However, these methods assume the crowd is evenly distributed; in reality crowd distribution is quite irregular. The Bayesian loss method [31] uses a Gaussian to construct a likelihood function for each annotated dot. However, it may not predict a correct density because the loss is underdetermined. Detailed analysis can be found in Sec 4.2.

### 2.2 Optimal Transport

We propose a novel loss function based on Optimal Transport (OT) [46]. For a better understanding of the proposed method, we briefly review the Monge-Kantorovich OT formulation in this section.

Optimal Transport refers to the optimal cost to transform one probability distribution to another. Let $\mathcal{X} = \{\mathbf{x}_i | \mathbf{x}_i \in \mathbb{R}^d\}_{i=1}^n$ and $\mathcal{Y} = \{\mathbf{y}_j | \mathbf{y}_j \in \mathbb{R}^d\}_{j=1}^n$ be two sets of points on $d$-dimensional vector space. Let $\boldsymbol{\mu}$ and $\boldsymbol{\nu}$ be two probability measures defined on $\mathcal{X}$ and $\mathcal{Y}$, respectively; $\boldsymbol{\mu}, \boldsymbol{\nu} \in \mathbb{R}_+^n$ and $\mathbf{1}_n^T \boldsymbol{\mu} = \mathbf{1}_n^T \boldsymbol{\nu} = 1$ ($\mathbf{1}_n$ is a $n$-dimensional vector of all ones). Let $c : \mathcal{X} \times \mathcal{Y} \mapsto \mathbb{R}_+$ be the cost function for moving from a point in $\mathcal{X}$ to a point in $\mathcal{Y}$, and $\mathbf{C}$ be the corresponding $n \times n$ cost matrix for the two sets of points: $\mathbf{C}_{ij} = c(\mathbf{x}_i, \mathbf{y}_j)$. Let $\Gamma$ be the set of all possible ways to transport probability mass from $\mathcal{X}$ to $\mathcal{Y}$: $\Gamma = \{\gamma \in \mathbb{R}_+^{n \times n} : \gamma \mathbf{1} = \boldsymbol{\mu}, \gamma^T \mathbf{1} = \boldsymbol{\nu}\}$. The Monge-Kantorovich's Optimal Transport (OT) cost between $\boldsymbol{\mu}$ and $\boldsymbol{\nu}$ is defined as:

$$\mathcal{W}(\boldsymbol{\mu}, \boldsymbol{\nu}) = \min_{\gamma \in \Gamma} \ \langle \mathbf{C}, \gamma \rangle. \tag{1}$$

Intuitively, if the probability distribution $\boldsymbol{\mu}$ is viewed as a unit amount of "dirt" piled on $\mathcal{X}$ and $\boldsymbol{\nu}$ a unit amount of dirt piled on $\mathcal{Y}$, the OT cost is the minimum "cost" of turning one pile into the other. The OT cost is a principal measurement to quantify the dissimilarity between two probability distributions, also taking into account the distance between "dirt" locations.

The OT cost can also be computed via the dual formulation:

$$\mathcal{W}(\boldsymbol{\mu}, \boldsymbol{\nu}) = \max_{\boldsymbol{\alpha}, \boldsymbol{\beta} \in \mathbb{R}^n} \langle \boldsymbol{\alpha}, \boldsymbol{\mu} \rangle + \langle \boldsymbol{\beta}, \boldsymbol{\nu} \rangle, \quad \text{s.t. } \alpha_i + \beta_j \leq c(\mathbf{x}_i, \mathbf{y}_j), \ \forall i, j. \tag{2}$$

## 3   DM-Count: Distribution Matching for Crowd Counting

We consider crowd counting as a distribution matching problem. In this section, we propose DM-Count: Distribution matching for crowd counting. A network for crowd counting inputs an image and outputs a map of density values. The final count estimate can be obtained by summing over the predicted density map. DM-Count is agnostic to different network architectures. In our experiments, we use the same network as in the Bayesian loss paper [31]. Unlike all previous density map estimation methods which need to use Gaussians to smooth ground truth annotations, DM-Count does not need any Gaussian to preprocess ground truth annotations.

Let $\mathbf{z} \in \mathbb{R}_+^n$ denote the vectorized binary map for dot-annotation and $\hat{\mathbf{z}} \in \mathbb{R}_+^n$ the vectorized predicted density map returned by a neural network. By viewing $\mathbf{z}$ and $\hat{\mathbf{z}}$ as unnormalized density functions, we formulate the loss function in DM-Count using three terms: the counting loss, the OT loss, and the Total Variation (TV) loss. The first term measures the difference between the total masses, while the last two measures the difference between the distributions of the normalized density functions.

**The Counting Loss**. Let $\| \cdot \|_1$ denote the $L_1$ norm of a vector, and so $\|\mathbf{z}\|_1, \|\hat{\mathbf{z}}\|_1$ are the ground truth and predicted counts respectively. The goal of crowd counting is to make $\|\hat{\mathbf{z}}\|_1$ as close as possible to $\|\mathbf{z}\|_1$, and the counting loss is defined as the absolute difference between them:

$$\ell_C(\mathbf{z}, \hat{\mathbf{z}}) = |\|\mathbf{z}\|_1 - \|\hat{\mathbf{z}}\|_1|. \tag{3}$$

**The Optimal Transport Loss**. Both $\mathbf{z}$ and $\hat{\mathbf{z}}$ are unnormalized density functions, but we can turn them into probability density functions (pdfs) by dividing them by the their respective total mass. Apart from OT, the Kullback-Leibler divergence and Jensen-Shannon divergence can also measure the similarity between two pdfs. However, these measurements do not provide valid gradients to train a network if the source distribution does not overlap with the target distribution [32]. Therefore, we propose the use of OT in this work. We define the OT loss as follows:

$$\ell_{OT}(\mathbf{z}, \hat{\mathbf{z}}) = \mathcal{W}\left(\frac{\mathbf{z}}{\|\mathbf{z}\|_1}, \frac{\hat{\mathbf{z}}}{\|\hat{\mathbf{z}}\|_1}\right) = \left\langle \boldsymbol{\alpha}^*, \frac{\mathbf{z}}{\|\mathbf{z}\|_1} \right\rangle + \left\langle \boldsymbol{\beta}^*, \frac{\hat{\mathbf{z}}}{\|\hat{\mathbf{z}}\|_1} \right\rangle, \tag{4}$$

where $\boldsymbol{\alpha}^*$ and $\boldsymbol{\beta}^*$ are the solutions of Problem (2). We use the quadratic transport cost, i.e., $c(\mathbf{z}(i), \hat{\mathbf{z}}(j)) = \|\mathbf{z}(i) - \hat{\mathbf{z}}(j)\|_2^2$, where $\mathbf{z}(i)$ and $\hat{\mathbf{z}}(j)$ are 2D coordinates of locations $i$ and $j$, respectively. To avoid the division-by-zero error, we add a machine precision to the denominator.

Since the entries in $\hat{\mathbf{z}}$ are non-negative, the gradient of Eq. (4) with respect to $\hat{\mathbf{z}}$ is:

$$\frac{\partial \ell_{OT}(\mathbf{z}, \hat{\mathbf{z}})}{\partial \hat{\mathbf{z}}} = \frac{\boldsymbol{\beta}^*}{\|\hat{\mathbf{z}}\|_1} - \frac{\langle \boldsymbol{\beta}^*, \hat{\mathbf{z}} \rangle}{\|\hat{\mathbf{z}}\|_1^2}. \tag{5}$$

This gradient can be back-propagated to learn the parameters of the density estimation network.

**Total Variation Loss**. In each training iteration, we use the Sinkhorn algorithm [34] to approximate $\boldsymbol{\alpha}^*$ and $\boldsymbol{\beta}^*$. The time complexity is $O(n^2 \log n / \epsilon^2)$ [9], where $\epsilon$ is the desired optimality gap, i.e., the upper bound for the difference between the returned objective and the optimal objective. When optimizing with the Sinkhorn algorithm, the objective decreases dramatically at the beginning but only converges slowly to the optimal objective in later iterations. In practice, we set the maximum number of iterations, and the Sinkhorn algorithm only returns an approximate solution. As a result, when we optimize the OT loss with the Sinkhorn algorithm, the predicted density map ends up close to the ground truth density map, but not exactly the same. The OT loss will approximate well the dense areas of the crowd, but the approximation might be poorer for the low density areas of the crowd. To address this issue, we additionally use the Total Variation (TV) loss, defined as[1]:

$$\ell_{TV}(\mathbf{z}, \hat{\mathbf{z}}) = \left\| \frac{\mathbf{z}}{\|\mathbf{z}\|_1} - \frac{\hat{\mathbf{z}}}{\|\hat{\mathbf{z}}\|_1} \right\|_{TV} = \frac{1}{2} \left\| \frac{\mathbf{z}}{\|\mathbf{z}\|_1} - \frac{\hat{\mathbf{z}}}{\|\hat{\mathbf{z}}\|_1} \right\|_1. \qquad (6)$$

The TV loss will also increase the stability of the training procedure. Optimizing the OT loss with the Sinkhorn algorithm is a min-max saddle point optimization procedure, which is similar to GAN optimization [13]. The stability of GAN training can be increased by adding a reconstruction loss, as shown in the Pix2Pix GAN [16]. To this end, the TV loss is similar to the reconstruction loss, and also increases the stability of the training procedure.

The gradient of the TV loss with respect to the predicted density map $\hat{\mathbf{z}}$ is:

$$\frac{\partial \ell_{TV}(\mathbf{z}, \hat{\mathbf{z}})}{\partial \hat{\mathbf{z}}} = -\frac{1}{2} \left( \frac{\text{sign}(\mathbf{v})}{\|\hat{\mathbf{z}}\|_1} - \frac{\langle \text{sign}(\mathbf{v}), \hat{\mathbf{z}} \rangle}{\|\hat{\mathbf{z}}\|_1^2} \right), \qquad (7)$$

where $\mathbf{v} = \mathbf{z}/\|\mathbf{z}\|_1 - \hat{\mathbf{z}}/\|\hat{\mathbf{z}}\|_1$, and $\text{sign}(\cdot)$ is the Sign function on each element of a vector.

**The Overall Objective**. The overall loss function is the combination of the counting loss, the OT loss, and the TV loss:

$$\ell(\mathbf{z}, \hat{\mathbf{z}}) = \ell_C(\mathbf{z}, \hat{\mathbf{z}}) + \lambda_1 \ell_{OT}(\mathbf{z}, \hat{\mathbf{z}}) + \lambda_2 \|\mathbf{z}\|_1 \ell_{TV}(\mathbf{z}, \hat{\mathbf{z}}), \qquad (8)$$

where $\lambda_1$ and $\lambda_2$ are tunable hyper-parameters for the OT and TV losses. To ensure that the TV loss has the same scale as the counting loss, we multiply this loss term with the total count.

Given $K$ training images $\{I_k\}_{k=1}^K$ with corresponding dot annotation maps $\{\mathbf{z}_k\}_{k=1}^K$, we will learn a deep neural network $f$ for density map estimation by minimizing: $L(f) = \frac{1}{K} \sum_{k=1}^K \ell(\mathbf{z}_k, f(I_k))$.

# 4 Generalization Bounds and Theoretical Analysis

In this section, we analyze the theoretical properties of the Gaussian smoothed methods, the Bayesian loss, and the proposed DM-Count. The proofs of the theorems in this section can be found in the supplementary material. First, we introduce some notations below.

Let $\mathcal{I}$ denote the set of images and $\mathcal{Z}$ the set of dot annotation maps. Let $\mathcal{D} = \{(I, \mathbf{z})\}$ be the joint distribution of crowd images and corresponding dot annotation maps. Let $\mathcal{H}$ be a hypothesis space. Each $h \in \mathcal{H}$ maps from $I \in \mathcal{I}$ to each dimension of $\mathbf{z} \in \mathcal{Z}$. Let $\mathcal{F} = \mathcal{H} \times \cdots \times \mathcal{H}$ ($n$ times) be the mapping space. Each $f \in \mathcal{F}$ maps $I \in \mathcal{I}$ to $\mathbf{z} \in \mathcal{Z}$. Let $\mathbf{t}$ be the Gaussian smoothed density map of each $\mathbf{z} \in \mathcal{D}$, and let $\tilde{\mathcal{D}} = \{(I, \mathbf{t})\}$ be the joint distribution of $(I, \mathbf{t})$. Let $S = \{(I_k, \mathbf{z}_k)\}_{k=1}^K$, and $\tilde{S} = \{(I_k, \mathbf{t}_k)\}_{k=1}^K$ be the finite sets of $K$ samples i.i.d. sampled from $\mathcal{D}$ and $\tilde{\mathcal{D}}$, respectively. Let $R_S(\mathcal{H})$ denote the empirical Rademacher complexity [3] for $\mathcal{H}$ w.r.t $S$. Given a data set $D \in \{\mathcal{D}, S, \tilde{\mathcal{D}}, \tilde{S}\}$, a mapping $f \in \mathcal{F}$ and a loss function $\ell$, let $\mathcal{R}(D, f, \ell) = \mathbb{E}_{(I, \mathbf{s}) \sim D}[\ell(\mathbf{s}, f(I))]$ denote the expected risk. Let $\ell_1(\mathbf{z}, \hat{\mathbf{z}}) = \|\mathbf{z} - \hat{\mathbf{z}}\|_1$. Let $f_\Delta^D = \operatorname{argmin}_{f \in \mathcal{F}} \mathcal{R}(D, f, \ell_\Delta)$ be the minimizer of $\mathcal{R}(D, f, \ell_\Delta)$ over a data set $D$ using the loss $\ell_\Delta$, where $D \in \{\mathcal{D}, S, \tilde{\mathcal{D}}, \tilde{S}\}$, and $\Delta \in \{1, C, OT, TV, \emptyset\}$.

## 4.1 Generalization Error Bounds of Gaussian Smoothed Methods

Many existing methods (e.g., [60, 20, 35]) use Gaussian-smoothed annotation maps for training. Below we give generalization error bounds when using the $\ell_1$ loss on the density maps.

**Theorem 1** *Assume that $\forall f \in \mathcal{F}$ and $(I, \mathbf{t}) \sim \tilde{\mathcal{D}}$, we have $\ell(\mathbf{t}, f(I)) \leq B$. Then, for any $0 < \delta < 1$, with probability of at least $1 - \delta$,*
*a) the upper bound of the generalization error is*

$$\mathcal{R}(\mathcal{D}, f_1^{\tilde{S}}, \ell_1) \leq \mathcal{R}(\tilde{\mathcal{D}}, f_1^{\tilde{\mathcal{D}}}, \ell_1) + 2n R_{\tilde{S}}(\mathcal{H}) + 5B\sqrt{2 \log{(8/\delta)}/K} + \mathbb{E}_{(I, \mathbf{z}) \sim \mathcal{D}} \|\mathbf{z} - \mathbf{t}\|_1,$$

*b) the lower bound of the generalization error is*

$$\mathcal{R}(\mathcal{D}, f_1^{\tilde{S}}, \ell_1) \geq \left| \mathbb{E}_{(I, \mathbf{z}) \sim \mathcal{D}} \|\mathbf{z} - \mathbf{t}\|_1 - \mathcal{R}(\tilde{\mathcal{D}}, f_1^{\tilde{S}}, \ell_1) \right|.$$

In this theorem, as the number of samples $K$ grows to infinity, $2n R_{\tilde{S}}(\mathcal{H})$ and $5B\sqrt{2 \log{(8/\delta)}/K}$ decrease to 0. Theorem 1.a) shows that the upper bound (worst case) of the expected risk $\mathcal{R}(\mathcal{D}, f_1^{\tilde{S}}, \ell_1)$, which is evaluated on real ground truth data using an empirical minimizer trained on the Gaussian smoothed ground truth, does not exceed $\mathcal{R}(\tilde{\mathcal{D}}, f_1^{\tilde{\mathcal{D}}}, \ell_1) + \mathbb{E}_{(I, \mathbf{z}) \sim \mathcal{D}} \|\mathbf{z} - \mathbf{t}\|_1$ given sufficient training data. Theorem 1.b) shows that the lower bound (best case) of $\mathcal{R}(\mathcal{D}, f_1^{\tilde{S}}, \ell_1)$ is not smaller than $|\mathbb{E}_{(I, \mathbf{z}) \sim \mathcal{D}} \|\mathbf{z} - \mathbf{t}\|_1 - \mathcal{R}(\tilde{\mathcal{D}}, f_1^{\tilde{S}}, \ell_1)|$. This means that if $\mathcal{R}(\tilde{\mathcal{D}}, f_1^{\tilde{S}}, \ell_1) \leq \mathbb{E}_{(I, \mathbf{z}) \sim \mathcal{D}} \|\mathbf{z} - \mathbf{t}\|_1$, then the smaller $\mathcal{R}(\tilde{\mathcal{D}}, f_1^{\tilde{S}}, \ell_1)$ is, the larger the expected risk $\mathcal{R}(\mathcal{D}, f_1^{\tilde{S}}, \ell_1)$ will be. In other words, the better a good model $f_1^{\tilde{S}}$ performs on the Gaussian smoothed ground truth $\tilde{\mathcal{D}}$, the poorer it generalizes on the real ground truth $\mathcal{D}$. Furthermore, as long as $\mathcal{R}(\tilde{\mathcal{D}}, f_1^{\tilde{S}}, \ell_1) \neq \mathbb{E}_{(I, \mathbf{z}) \sim \mathcal{D}} \|\mathbf{z} - \mathbf{t}\|_1$, we have $\mathcal{R}(\mathcal{D}, f_1^{\tilde{S}}, \ell_1) > 0$. $\mathcal{R}(\mathcal{D}, f_1^{\tilde{S}}, \ell_1)$ can be as large as $\mathbb{E}_{(I, \mathbf{z}) \sim \mathcal{D}} \|\mathbf{z} - \mathbf{t}\|_1$ when $\mathcal{R}(\tilde{\mathcal{D}}, f_1^{\tilde{S}}, \ell_1) = 0$. This is undesirable because we want the risk $\mathcal{R}(\mathcal{D}, f_1^{\tilde{S}}, \ell_1)$ evaluated on the real ground truth to be 0 as well.

## 4.2 The Underdetermined Bayesian Loss

The Bayesian Loss [31] is:

$$\ell_{Bayesian}(\mathbf{z}, \hat{\mathbf{z}}) = \sum_{i=1}^{N} |1 - \langle \mathbf{p}_i, \hat{\mathbf{z}} \rangle|, \quad \text{where} \quad \mathbf{p}_i = \frac{\mathcal{N}(\mathbf{q}_i, \sigma^2 \mathbf{1}_{2 \times 2})}{\sum_{i=1}^{N} \mathcal{N}(\mathbf{q}_i, \sigma^2 \mathbf{1}_{2 \times 2})}, \tag{9}$$

and $N$ is number of people of $\mathbf{z}$, and $\mathcal{N}(\mathbf{q}_i, \sigma^2 \mathbf{1}_{2 \times 2})$ is a Gaussian distribution centered at $\mathbf{q}_i$ with variance $\sigma^2 \mathbf{1}_{2 \times 2}$. $\mathbf{q}_i$ is the $i^{th}$ annotated dot in $\mathbf{z}$. The dimension of $\mathbf{p}_i$ and $\mathbf{z}$ is $n$, the number of pixels of the density map. However, since the number of annotated dots $N$ is less than $n$, the Bayesian loss is underdetermined. For a ground truth annotation $\mathbf{z}$, there are infinitely many $\hat{\mathbf{z}}$ with $\ell_{Bayesian}(\mathbf{z}, \hat{\mathbf{z}}) = 0$ and $\hat{\mathbf{z}} \neq \mathbf{z}$. Therefore, the predicted density map could be very different from the ground truth density map.

## 4.3 The Generalization Error Bounds of the Losses in DM-Count

We give the generalization error bounds of the losses in the proposed method in the following theorem.

**Theorem 2** *Assume that $\forall f \in \mathcal{F}$ and $(I, \mathbf{z}) \sim \mathcal{D}$, we have $\|\mathbf{z}\|_1 \geq 1$, $\|f(I)\|_1 \geq 1$ (can be satisfied by adding a dummy dimension with value of 1 to both $\mathbf{z}$ and $f(I)$) and $\ell_C(\mathbf{z}, f(I)) \leq B$. Then, for any $0 < \delta < 1$, with probability of at least $1 - \delta$*
*a) the generalization error bound of the counting loss is*

$$\mathcal{R}(\mathcal{D}, f_C^S, \ell_C) \leq \mathcal{R}(\mathcal{D}, f_C^{\mathcal{D}}, \ell_C) + 2n R_S(\mathcal{H}) + 5B\sqrt{2 \log{(8/\delta)}/K},$$

*b) the generalization error bound of the OT loss is*

$$\mathcal{R}(\mathcal{D}, f_{OT}^S, \ell_{OT}) \leq \mathcal{R}(\mathcal{D}, f_{OT}^{\mathcal{D}}, \ell_{OT}) + 4\mathbf{C}_\infty n^2 R_S(\mathcal{H}) + 5\mathbf{C}_\infty \sqrt{2 \log{(8/\delta)}/K},$$

*c) the generalization error bound of the TV loss is*

$$\mathcal{R}(\mathcal{D}, f_{TV}^S, \ell_{TV}) \leq \mathcal{R}(\mathcal{D}, f_{TV}^{\mathcal{D}}, \ell_{TV}) + n^2 R_S(\mathcal{H}) + 5\sqrt{2 \log{(8/\delta)}/K},$$

*d) the generalization error bound of the overall loss is*

$$\mathcal{R}(\mathcal{D}, f^S, \ell) \leq \mathcal{R}(\mathcal{D}, f^{\mathcal{D}}, \ell) + (2n + 4\lambda_1 \mathbf{C}_\infty n^2 + \lambda_2 N n^2) R_S(\mathcal{H})$$
$$+ 5(B + \lambda_1 \mathbf{C}_\infty + \lambda_2 N)\sqrt{2 \log{(8/\delta)}/K},$$

*where $\mathbf{C}_\infty$ is the maximum cost in the cost matrix in OT, and $N = \sup\{\|\mathbf{z}\|_1 \mid \forall (I, \mathbf{z}) \sim \mathcal{D}\}$ is the maximum count number over a dataset.*

| Target density map | Pixel-wise loss | Bayesian loss | DM-Count (proposed) |
|:---:|:---:|:---:|:---:|

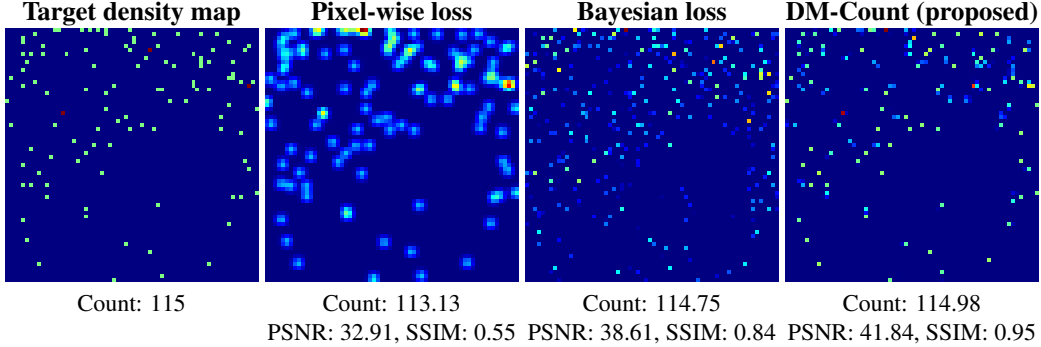

| Count: 115 | Count: 113.13 | Count: 114.75 | Count: 114.98 |
|:---:|:---:|:---:|:---:|
| | PSNR: 32.91, SSIM: 0.55 | PSNR: 38.61, SSIM: 0.84 | PSNR: 41.84, SSIM: 0.95 |

Figure 1: **Comparison of different methods on toy data.** The pixel-wise loss generates a blurry density map with a higher counting error. The Bayesian loss produces dissimilar density maps from the ground truth, with high values in many locations with no annotations. DM-Count is able to produce more accurate crowd count and localization than the other two methods.

In the above theorem, as $K$ grows, $R_S(\mathcal{H})$ and $\sqrt{2\log(1/\delta)/K}$ decrease. All the expected risks $\mathcal{R}(\mathcal{D}, f_\Delta^S, \ell_\Delta)$ using the empirical minimizers $f_\Delta^S$ converge to the expected risks $\mathcal{R}(\mathcal{D}, f_\Delta^{\mathcal{D}}, \ell_\Delta)$, $\Delta \in \{C, OT, TV, \emptyset\}$ using optimal minimizers $f_\Delta^{\mathcal{D}}$. This means that all the upper bounds are tight. In addition, all upper bounds are tighter than the upper bound of the Gaussian smoothed methods shown in Theorem 1.a). The bound of the OT loss in Theorem 2.b) is related to the maximum transport cost $\mathbf{C}_\infty$. Therefore, we need to use a smaller transport cost in OT for better generalization performance. The coefficient of $R_S(\mathcal{H})$ for the counting loss is $O(n)$, and for the OT loss and the TV loss is $O(n^2)$. This means that for larger image size, we need more images to train. The number is linear to the size of $\mathbf{z}$ using solely the counting loss, and quadratic using solely the OT loss or the TV loss. When using all three losses, we need to set $\lambda_1$ and $\lambda_2$ to be small in order to balance the three losses.

## 5 Experiments

In this section, we describe experiments on toy data and on benchmark crowd counting datasets. More detailed dataset descriptions, implementation details and experimental settings can be found in the supplementary material.

### 5.1 Results on Toy Data

To understand the empirical behavior of different methods, we consider a toy problem where the task is to move a source density map $\hat{\mathbf{z}}$ to a target density map $\mathbf{z}$ using the Pixel-wise loss, the Bayesian loss and DM-Count. The source density map $\hat{\mathbf{z}}$ is initialized from a uniform distribution between 0 and 0.01, and the target density map is shown in the leftmost figure in Fig. 1. All three methods start from the same source density map. Fig. 1 visualizes the final $\hat{\mathbf{z}}$ at convergence. The Pixel-wise loss yields a blurry density map with a higher count. The Bayesian loss performs better than the Pixel-wise loss in terms of counting error, Peak Signal-to-Noise Ratio (PSNR) and Structural Similarity in Image (SSIM) [52], but the resulting density map is quite different from the target, with high values at many locations where no dots are annotated. This confirms our analysis that the Bayesian loss corresponds to an underdetermined system such that the output density map could be very different from the target density map. In contrast, DM-Count is able to produce a more accurate count and density map. DM-Count outperforms the Bayesian loss by a large margin in both PSNR and SSIM.

### 5.2 Results on Benchmark Datasets

We perform experiments on four challenging crowd counting datasets: UCF-QNRF [15], NWPU [51], ShanghaiTech [60], and UCF-CC-50 [14]. It is worth noting that the **NWPU** dataset is the largest-scale and most challenging crowd counting dataset publicly available today. The ground truth counts for test images are not released, and the results on the test set must be obtained by submitting to the evaluation server at `https://www.crowdbenchmark.com/nwpucrowd.html`. Following previous work [35, 15, 4, 14, 60], we use the following metrics: Mean Absolute Error (MAE), Root Mean Squared Error (RMSE), and Mean Normalized Absolute Error (NAE) as evaluation metrics. For

|  | UCF-QNRF | | ShanghaiTech A | | ShanghaiTech B | | UCF-CC-50 | |
|---|---|---|---|---|---|---|---|---|
|  | MAE | RMSE | MAE | RMSE | MAE | RMSE | MAE | RMSE |
| Crowd CNN [58] | - | - | 181.8 | 277.7 | 32.0 | 49.8 | 467.0 | 498.5 |
| MCNN [60] | 277 | 426 | 110.2 | 173.2 | 26.4 | 41.3 | 377.6 | 509.1 |
| CMTL [41] | 252 | 514 | 101.3 | 152.4 | 20.0 | 31.1 | 322.8 | 341.4 |
| Switch CNN [2] | 228 | 445 | 90.4 | 135.0 | 21.6 | 33.4 | 318.1 | 439.2 |
| IG-CNN [1] | - | - | 72.5 | 118.2 | 13.6 | 21.1 | 291.4 | 349.4 |
| ic-CNN [35] | - | - | 68.5 | 116.2 | 10.7 | 16.0 | 260.9 | 365.5 |
| CSR Net [20] | - | - | 68.2 | 115.0 | 10.6 | 16.0 | 266.1 | 397.5 |
| SANet [4] | - | - | 67.0 | 104.5 | 8.4 | 13.6 | 258.4 | 334.9 |
| CL-CNN [15] | 132 | 191 | - | - | - | - | - | - |
| PACNN [40] | - | - | 62.4 | 102.0 | 7.6 | 11.8 | 241.7 | 320.7 |
| CAN [27] | 107 | 183 | 62.3 | 100.0 | 7.8 | 12.2 | 212.2 | **243.7** |
| SFCN [50] | 102 | 171 | 64.8 | 107.5 | 7.6 | 13.0 | 214.2 | 318.2 |
| ANF [57] | 110 | 174 | 63.9 | 99.4 | 8.3 | 13.2 | 250.2 | 340.0 |
| Wan *et al.* [47] | 101 | 176 | 64.7 | 97.1 | 8.1 | 13.6 | - | - |
| Pixel-wise Loss [31] | 106.8 | 183.7 | 68.6 | 110.1 | 8.5 | 13.9 | 251.6 | 331.3 |
| Bayesian Loss [31] | 88.7 | 154.8 | 62.8 | 101.8 | 7.7 | 12.7 | 229.3 | 308.2 |
| DM-Count (proposed) | **85.6** | **148.3** | **59.7** | **95.7** | **7.4** | **11.8** | **211.0** | 291.5 |

Table 1: **Results on the UCF-QNRF, Shanghai Tech, and UCF-CC-50 datasets**.

|  | Backbone | Validation set | | Test set | | |
|---|---|---|---|---|---|---|
|  |  | MAE | RMSE | MAE | RMSE | NAE |
| MCNN [60] | FS | 218.5 | 700.6 | 232.5 | 714.6 | 1.063 |
| CSR net [20] | VGG-16 | 104.8 | 433.4 | 121.3 | 387.8 | 0.604 |
| PCC-Net-VGG [10] | VGG-16 | 100.7 | 573.1 | 112.3 | 457.0 | 0.251 |
| CAN [27] | VGG-16 | 93.5 | 489.9 | 106.3 | **386.5** | 0.295 |
| SCAR [11] | VGG-16 | 81.5 | 397.9 | 110.0 | 495.3 | 0.288 |
| Bayesian Loss [31] | VGG-19 | 93.6 | 470.3 | 105.4 | 454.2 | 0.203 |
| SFCN [50] | ResNet-101 | 95.4 | 608.3 | 105.7 | 424.1 | 0.254 |
| DM-Count (proposed) | VGG-19 | **70.5** | **357.6** | **88.4** | 388.6 | **0.169** |

Table 2: **Results of various methods on the NWPU validation and test sets**.

all three metrics, the smaller the better. For a fair comparison, we use the same network as in the Bayesian loss paper [31]. In all experiments, we set $\lambda_1 = 0.1, \lambda_2 = 0.01$, and the Sinkhorn entropic regularization parameter to 10. The number of Sinkhorn iterations is set to 100. On average, the OT computation time is 25*ms* for each image.

**Quantitative Results**. Tables 1 and 2 compare the performance of DM-Count against various methods. In all experiments, DM-Count outperforms all other methods except CAN under MSE in NWPU (where they are comparable). Although we use the same set of hyper-parameters for DM-Count in all experiments, DM-Count still achieves the best performance, suggesting that DM-Count's performance is stable across various datasets.

DM-Count outperforms the Pixel-wise loss and the Bayesian loss, when used in the same network architecture and training procedure as DM-Count, in all the experiments. This demonstrates the effectiveness of the proposed loss. The pixel-wise loss is much worse than DM-Count in Table 1. Additionally, even without using a multi-scale architecture as in [4, 47], or a deeper network as in [2, 50], DM-Count still achieves state-of-the-art performance on all four datasets. This indicates the importance of having a good loss function in crowd counting.

On the large-scale and challenging datasets UCF-QNRF and NWPU, DM-Count significantly outperforms the state-of-the-art methods. Specifically, on the UCF-QNRF dataset, DM-Count reduces the MAE and MSE of the Bayesian loss from 88.7 to 85.6 and from 154.8 to 148.3, respectively. Notably, on the NWPU test set (obtained by submitting to the evaluation server), DM-Count reduces the MAE and NAE by a large margin, from 105.4 to 88.4 in MAE and from 0.203 to 0.169 in NAE.

| Image | Pixel-wise loss | Bayesian loss | DM-Count (proposed) |
|---|---|---|---|

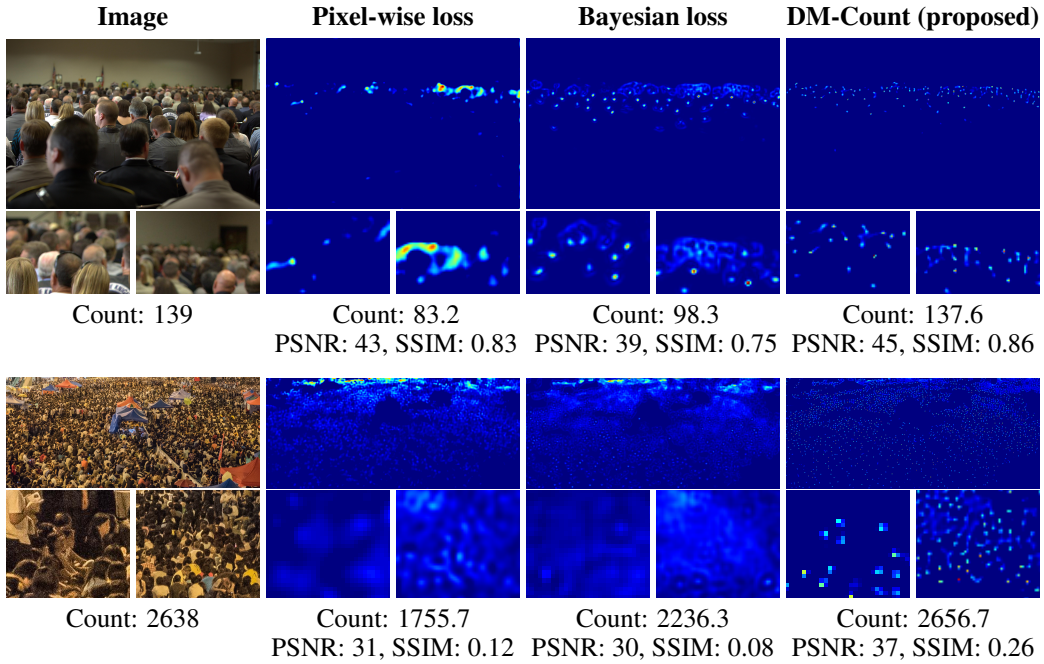

| Count: 139 | Count: 83.2 | Count: 98.3 | Count: 137.6 |
|---|---|---|---|
| | PSNR: 43, SSIM: 0.83 | PSNR: 39, SSIM: 0.75 | PSNR: 45, SSIM: 0.86 |
| Count: 2638 | Count: 1755.7 | Count: 2236.3 | Count: 2656.7 |
| | PSNR: 31, SSIM: 0.12 | PSNR: 30, SSIM: 0.08 | PSNR: 37, SSIM: 0.26 |

Figure 2: **Density map visualization.** Comparison between Pixel-wise loss, Bayesian loss and DM-Count. The pixel-wise and Bayesian losses fail to localize people well in dense regions. DM-Count is able to localize people both in dense and sparse regions. The Count number, PSNR and SSIM metrics suggest that DM-Count produces more accurate count numbers and better density maps.

| Image | DM-Count Predicted Result |
|---|---|

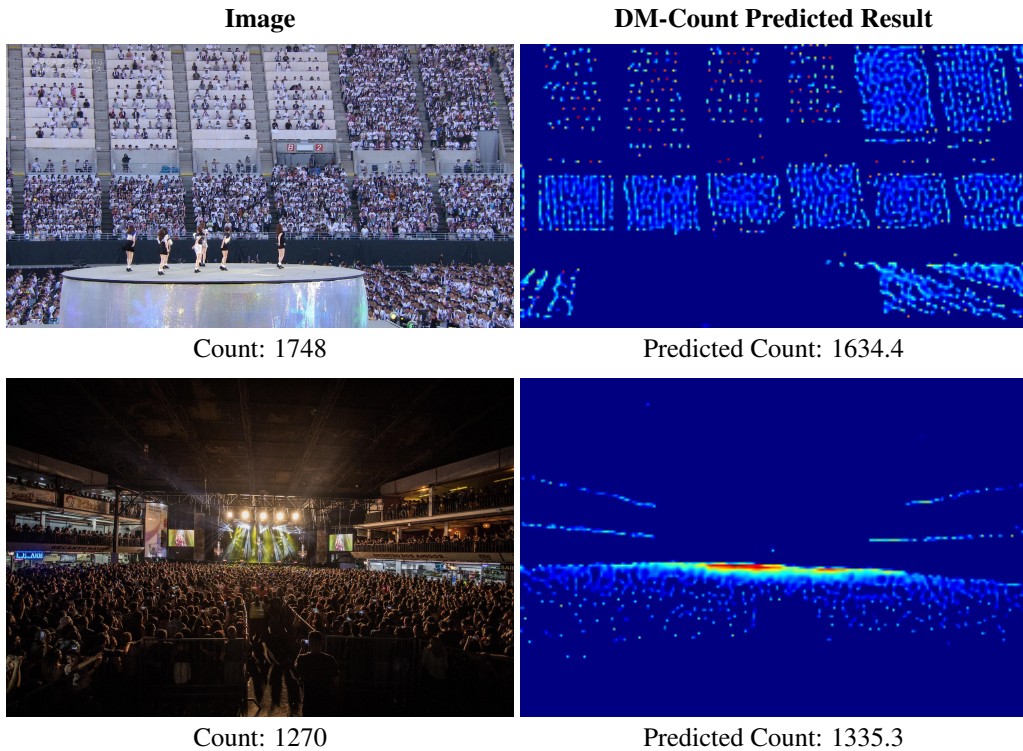

| Count: 1748 | Predicted Count: 1634.4 |
|---|---|
| Count: 1270 | Predicted Count: 1335.3 |

Figure 3: **Density map visualization on the NWPU validation set.**

**Qualitative Results**. Fig. 2 shows the predicted density maps of the Pixel-wise loss, the Bayesian loss and DM-Count. This figure demonstrates that: 1) DM-Count produces count numbers that are closer to the ground truth numbers, 2) DM-Count produces much sharper density maps than the

| # Sinkhorn Iters | MAE | RMSE |
|---|---|---|
| 50 | 90.8 | 162.1 |
| 100 | 85.6 | 148.3 |
| 120 | 85.5 | 151.5 |

Table 3: **Effect of # of Sinkhorn iterations.**

| Method | MAE | RMSE |
|---|---|---|
| Pixel-wise loss | 144.1 | 232.5 |
| Bayesian loss | 108.4 | 187.2 |
| DM-Count | **105.6** | **181.6** |

Table 4: **Robustness to noisy annotations.**

Pixel-wise and Bayesian losses. In Fig. 2, DM-Count produces much higher PSNRs and SSIMs than the Pixel-wise and Bayesian losses. The average PSNR and SSIM over the whole UCF-QNRF test set for the Pixel-wise loss are 34.79 and 0.43, for the Bayesian loss are 34.55 and 0.42, and for DM-Count are 40.65 and 0.55, respectively. Because the Pixel-wise loss uses the Gaussian smoothed ground truth, it produces blurrier density maps than the real ground truth. This empirically verifies our theoretical analysis of the generalization bound of Gaussian smoothed methods. As shown in the figure, the Pixel-wise and Bayesian losses are unable to localize people in dense regions. In contrast, DM-Count localizes people well in both dense and sparse regions. Fig. 3 shows predicted density maps by DM-Count. The predicted density maps correspond well to crowd densities in both sparse and dense areas, demonstrating the effectiveness of DM-Count in spatial density estimation.

## 5.3 Ablation Studies

**Hyper-parameter study**. We tune $\lambda_1$ and $\lambda_2$ in DM-Count on the UCF-QNRF dataset. First, we fix $\lambda_1$ to 0.1 and tune $\lambda_2$ from 0.01, 0.05 to 0.1. The MAE varies from 85.6, 87.8 to 88.5. As $\lambda_2 = 0.01$ achieves the best result, we fix $\lambda_2$ to 0.01 and tune $\lambda_1$ from 0.01, 0.05 to 0.1. The MAE varies from 87.2, 86.2 to 85.6. Thus, we set $\lambda_1 = 0.1$, $\lambda_2 = 0.01$ and use them on all the datasets.

**Effect of the number of Sinkhorn iterations**. Table 3 lists the results of DM-Count on the UCF-QNRF dataset using different numbers of Sinkhorn iterations. As shown in this table, using a small number of iterations lowers the performance of DM-Count, which indicates that we obtain inaccurate OT solutions. When the number of iterations increases to 100, DM-Count outperforms the previous state-of-the-art. The performance plateaued after the number of iterations crossed 100. Therefore, in all of our experiments, we use 100 Sinkhorn iterations for DM-Count.

**Contribution of each component**. The loss in DM-Count is composed of three components, the counting loss, the OT loss and the TV loss. We study the contribution of each component on the UCF-QNRF dataset. Results are listed in Table 5. As seen in the Table, all components are essential to the final performance. However, the OT loss is the most important component.

| Component | Combinations | | | |
|---|---|---|---|---|
| Counting loss | ✓ | ✓ | ✓ | ✓ |
| OT loss | | | ✓ | ✓ |
| TV loss | | ✓ | | ✓ |
| MAE | 103.1 | 94.9 | 89.3 | 85.6 |
| RMSE | 175.9 | 167.4 | 161.3 | 148.3 |

Table 5: **Component analysis**

**Robustness to noisy annotations**. Crowd annotation is performed by placing a single dot on a person. Such process is ambiguous and could lead to inevitable annotation errors. We study how different loss functions perform w.r.t. annotation errors. We add uniform random noise to the original annotation and train different models with the same noisy annotation. The noise is randomly generated between 0 and 5% of the image height, and is about 80 pixels on average. As shown in Table 4, the proposed DM-Count is more robust to annotation errors compared to the pixel-wise Bayesian losses.

## 6 Conclusion

In this paper, we have shown that using the Gaussian kernel to smooth the ground truth dot annotations can hurt the generalization bound of a model when testing on the real ground truth data. Instead, we consider crowd counting as a distribution matching problem and propose DM-Count, based on Optimal Transport, to address this problem. Unlike prior work, DM-Count does not need a Gaussian kernel to smooth the annotated dots. The generalization error bound of DM-Count is tighter than that of the Gaussian smoothed methods. Extensive experiments on four crowd counting benchmarks demonstrated that DM-Count significantly outperforms previous state-of-the-art methods.

## Broader Impact

Our work is able to more accurately estimate the crowd size in images or videos, such that it can guide crowd control and improve public safety. The estimated crowd count results are interpretable, with better crowd localization, which will increase transparency of the results for critical applications. In an age when the size of the crowd in various political events often becomes a point of heated dispute, having transparent, accurate and objective counting methods could help the historical record, as well a public acceptance of the estimates. Our method could potentially be used to protect public health by monitoring social distancing which is becoming increasingly important during the current epidemic. This method does not leverage biases in the data. The proposed method for counting is general, with possible applications to biomedical cell counting, live stock counting and etc. Our work can be adapted to count moving crowds.

## Acknowledgments and Disclosure of Funding

This research was partially supported by US National Science Foundation Award IIS-1763981, the SUNY2020 Infrastructure Transportation Security Center, and Air Force Research Laboratory (AFRL) DARPA FA8750-19-2-1003, the Partner University Fund, and a gift from Adobe.

## Footnotes

[1] In the training loss context, Total Variation refers to the total variation distance of two probability measures. A formal definition can be found in [45, Definition 2.4, page 83]. Eq. (6) is [45, Lemma 2.1, page 84].

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
