[Supplementary Material]

# Distribution Matching for Crowd Counting
# Supplementary Material

**Boyu Wang**[*]  **Huidong Liu**[*]  **Dimitris Samaras**  **Minh Hoai**
Department of Computer Science, Stony Brook University, Stony Brook, NY 11790
{boywang, huidliu, samaras, minhhoai}@cs.stonybrook.edu
[*]indicates equal contribution

This document is structured as follows:

- In Section 1, we give the proofs of the theorems in the main paper.
- In Section 2, we give the detailed experimental settings, dataset descriptions, and implementation details. We also conduct some ablation studies on the number of Sinkhorn iterations used in DM-Count and investigate the robustness of different methods to noisy annotations.
- In Section 3, We show some more qualitative results of different methods and also show some examples where DM-Count does not perform well.
- Finally in Section 4 we show the leaderboard ranking record on the time of the main paper submission.

## 1 Proofs of Theorems 1 and 2 in the Main Paper

Let $\mathcal{I}$ denote the set of images and $\mathcal{Z}$ the set of dot annotation maps. Let $\mathcal{D} = \{(I, \mathbf{z})\}$ be the joint distribution of crowd images and corresponding dot annotation maps. Let $\mathcal{H}$ be a hypothesis space. Each $h \in \mathcal{H}$ maps from $I \in \mathcal{I}$ to each dimension of $\mathbf{z} \in \mathcal{Z}$. Let $\mathcal{F} = \mathcal{H} \times \cdots \times \mathcal{H}$ ($n$ times) be the mapping space. Each $f \in \mathcal{F}$ maps $I \in \mathcal{I}$ to $\mathbf{z} \in \mathcal{Z}$. Let $\mathbf{t}$ be the Gaussian smoothed density map of each $\mathbf{z} \in \mathcal{D}$, and let $\tilde{\mathcal{D}} = \{(I, \mathbf{t})\}$ be the joint distribution of $(I, \mathbf{t})$. Let $S = \{(I_k, \mathbf{z}_k)\}_{k=1}^{K}$, and $\tilde{S} = \{(I_k, \mathbf{t}_k)\}_{k=1}^{K}$ be the finite sets of $K$ samples i.i.d. sampled from $\mathcal{D}$ and $\tilde{\mathcal{D}}$, respectively. Let $R_S(\mathcal{H})$ denote the empirical Rademacher complexity [2] for $\mathcal{H}$ w.r.t $S$. Given a data set $D \in \{\mathcal{D}, S, \tilde{\mathcal{D}}, \tilde{S}\}$, a mapping $f \in \mathcal{F}$ and a loss function $\ell$, let $\mathcal{R}(D, f, \ell) = \mathbb{E}_{(I,\mathbf{s}) \sim D}[\ell(\mathbf{s}, f(I))]$ denote the expected risk. Let $\ell_1(\mathbf{z}, \hat{\mathbf{z}}) = \|\mathbf{z} - \hat{\mathbf{z}}\|_1$. Let $f_\Delta^D = \operatorname{argmin}_{f \in \mathcal{F}} \mathcal{R}(D, f, \ell_\Delta)$ be the minimizer of $\mathcal{R}(D, f, \ell_\Delta)$ over a data set $D$ using the loss $\ell_\Delta$, where $D \in \{\mathcal{D}, S, \tilde{\mathcal{D}}, \tilde{S}\}$, and $\Delta \in \{1, C, OT, TV, \emptyset\}$. Let $\operatorname{dom} f$ denote the domain of a mapping $f$. Denote the maximum value in a matrix $\mathbf{C}$ by $\mathbf{C}_\infty$.

First, we give two definitions that we will use in the proofs.

**Definition 1.** *(Empirical Rademacher complexity [2]) Let $\mathcal{G}$ be a space of functions mapping from $\mathcal{I}$ to $\mathbb{R}$. The empirical Rademacher complexity of $\mathcal{G}$ with respect to the sample $S$ is defined as:*

$$R_S(\mathcal{G}) = \mathbb{E}_{\boldsymbol{\sigma}} \left[ \sup_{g \in \mathcal{G}} \frac{1}{K} \sum_{k=1}^{K} \sigma_k g(I_k) \right], \tag{1}$$

*where $\boldsymbol{\sigma} = (\sigma_1, .., \sigma_K)^{\mathsf{T}}$, with $\sigma_k$ is independent uniform random variables taking values in $\{+1, -1\}$ the random variables $\sigma_k$ are called Rademacher variables. The Rademacher complexity is defined by taking the expectation with respect to the samples $S$,*

$$R_K(\mathcal{G}) = \mathbb{E}_S[R_S(\mathcal{G})]. \tag{2}$$

**Definition 2.** *A function $g : \mathbb{R}^m \mapsto \mathbb{R}$ is $L$-Lipschitz under the $L_1$ metric if $\forall \mathbf{a} \in \operatorname{dom} g$ and $\mathbf{b} \in \operatorname{dom} g$, we have*

$$|g(\mathbf{a}) - g(\mathbf{b})| \le L \|\mathbf{a} - \mathbf{b}\|_1. \tag{3}$$

Before proving the proposed theorems in the main paper, we state the following two theorems, introduced in [7, 9]. Theorem A provides us a tool for analyzing the Lipschitz constant under the $L_p$ metric, and Theorem B presents the generalization error bound in a general hypothesis space.

**Theorem A.** *[7]. For a Lipschitz function $g(\mathbf{x})$, $g : \mathbb{R}^m \mapsto \mathbb{R}$, we have*

$$|g(\mathbf{x}) - g(\mathbf{y})| \leq L_p \|\mathbf{x} - \mathbf{y}\|_q, \tag{4}$$

*where $L_p = \sup\{\|\nabla g(\mathbf{x})\|_p, \mathbf{x} \in \mathrm{dom}\, g\}$ is the Lipschitz constant, $\nabla g(\mathbf{x})$ is the gradient of function $g$ at $\mathbf{x}$, and $1/p + 1/q = 1$, $1 \leq p, q \leq \infty$.*

**Theorem B.** *[9] [Theorem 26.5, Chapter 26, Page 378]. Let $\mathcal{G}$ denote a space of functions. Each $g \in \mathcal{G}$ maps from $\mathcal{X}$ to $\mathbb{R}$. $\mathbf{x} \in \mathcal{X}$ has a distribution of $\mathcal{D}$. Let $S = \{\mathbf{x}_k\}_{k=1}^K$ be the set of samples i.i.d. sampled from $\mathcal{D}$. Let $L_{\mathcal{D}}(g) = \mathbb{E}_{\mathbf{x} \sim \mathcal{D}}[g(\mathbf{x})]$, and $L_S(g) = \mathbb{E}_{\mathbf{x} \sim S}[g(\mathbf{x})]$. Let $\hat{g} = \arg\min_g L_S(g)$ and $g^* = \arg\min_g L_{\mathcal{D}}(g)$. Assume for all $\mathbf{x} \sim \mathcal{D}$ and $g \in \mathcal{G}$ we have $|g(\mathbf{x})| \leq B$. Then,*

$$L_{\mathcal{D}}(\hat{g}) \leq L_{\mathcal{D}}(g^*) + 2R_S(\mathcal{G}) + 5B\sqrt{2\log(8/\delta)/K}. \tag{5}$$

We propose the following five lemmas which are essential for proving the proposed theorems. Lemmas A, B, C and D give the Lipschitz constants of different loss functions. Lemma E builds the connection between the Rademacher complexity of a hypothesis space and the Rademacher complexities of the space of feature (density map) extraction functions.

**Lemma A.** *Let $\mathbf{x}, \mathbf{x}' \in \mathbb{R}_+^n$, and $\ell(\mathbf{x}, \mathbf{x}') = \|\mathbf{x} - \mathbf{x}'\|_1$. Then, $\ell(\mathbf{x}, \mathbf{x}')$ is 1-Lipschitz w.r.t. $(\mathbf{x}, \mathbf{x}')$ under the $L_1$ metric, i.e., $\forall \mathbf{a}, \mathbf{a}', \mathbf{b}, \mathbf{b}' \in \mathbb{R}_+^n$, we have*

$$|\ell(\mathbf{a}, \mathbf{a}') - \ell(\mathbf{b}, \mathbf{b}')| \leq \|\mathbf{a} - \mathbf{b}\|_1 + \|\mathbf{a}' - \mathbf{b}'\|_1. \tag{6}$$

*Proof.* By the triangle inequality,

$$\begin{aligned}
|\ell(\mathbf{a}, \mathbf{a}') - \ell(\mathbf{b}, \mathbf{b}')| &= |\ell(\mathbf{a}, \mathbf{a}') - \ell(\mathbf{b}, \mathbf{a}') + \ell(\mathbf{b}, \mathbf{a}') - \ell(\mathbf{b}, \mathbf{b}')| \\
&\leq |\ell(\mathbf{a}, \mathbf{b})| + |\ell(\mathbf{a}', \mathbf{b}')| = \|\mathbf{a} - \mathbf{b}\|_1 + \|\mathbf{a}' - \mathbf{b}'\|_1. \tag{7}
\end{aligned}$$

$\square$

**Lemma B.** *Let $\mathbf{x}, \mathbf{x}' \in \mathbb{R}_+^n$, and $\ell(\mathbf{x}, \mathbf{x}') = |\|\mathbf{x}\|_1 - \|\mathbf{x}'\|_1|$. Then, $\ell(\mathbf{x}, \mathbf{x}')$ is 1-Lipschitz w.r.t. $(\mathbf{x}, \mathbf{x}')$ under the $L_1$ metric, i.e., $\forall \mathbf{a}, \mathbf{a}', \mathbf{b}, \mathbf{b}' \in \mathbb{R}_+^n$, we have*

$$|\ell(\mathbf{a}, \mathbf{a}') - \ell(\mathbf{b}, \mathbf{b}')| \leq \|\mathbf{a} - \mathbf{b}\|_1 + \|\mathbf{a}' - \mathbf{b}'\|_1. \tag{8}$$

*Proof.* As each element in $\mathbf{x}$ and $\mathbf{x}'$ is non-negative,

$$\ell(\mathbf{x}, \mathbf{x}') = |\mathbf{1}^\intercal \mathbf{x} - \mathbf{1}^\intercal \mathbf{x}'|, \tag{9}$$

where $\mathbf{1}$ is a vector of all ones. Then, computing the derivative of $\ell$ w.r.t. $(\mathbf{x}, \mathbf{x}')$:

$$\left\| \begin{bmatrix} \dfrac{\partial \ell(\mathbf{x}, \mathbf{x}')}{\mathbf{x}} \\[2mm] \dfrac{\partial \ell(\mathbf{x}, \mathbf{x}')}{\mathbf{x}'} \end{bmatrix} \right\|_\infty = \left\| \begin{bmatrix} \mathbf{1} \\ -\mathbf{1} \end{bmatrix} \right\|_\infty = 1. \tag{10}$$

By applying Theorem $A$ with $p = \infty$ and $q = 1$, we get that $\ell(\mathbf{x}, \mathbf{x}')$ is 1-Lipschitz under the $L_1$ metric. $\square$

**Lemma C.** *Let $\mathbf{x}, \mathbf{x}' \in \mathbb{R}_+^n$, and $\ell(\mathbf{x}, \mathbf{x}') = \|\mathbf{x}/\|\mathbf{x}\|_1 - \mathbf{x}'/\|\mathbf{x}'\|_1\|_1$. Assume that $\forall \mathbf{x}, \mathbf{x}'$, we have $\|\mathbf{x}\|_1 \geq 1$ and $\|\mathbf{x}'\|_1 \geq 1$. Then, $\ell(\mathbf{x}, \mathbf{x}')$ is $2n$-Lipschitz w.r.t. $(\mathbf{x}, \mathbf{x}')$ under the $L_1$ metric, i.e., $\forall \mathbf{a}, \mathbf{a}', \mathbf{b}, \mathbf{b}' \in \mathbb{R}_+^n$, we have*

$$|\ell(\mathbf{a}, \mathbf{a}') - \ell(\mathbf{b}, \mathbf{b}')| \leq 2n(\|\mathbf{a} - \mathbf{b}\|_1 + \|\mathbf{a}' - \mathbf{b}'\|_1). \tag{11}$$

*Proof.* Let $\delta : \mathbb{R}_+^n \mapsto \mathbb{R}_+^n$, and $\delta(\mathbf{x}) = \mathbf{x}/\|\mathbf{x}\|_1$. First we show that $\|\partial \delta_i(\mathbf{x})/\partial \mathbf{x}\|_2 \leq 2$.

$$\frac{\partial \delta_i(\mathbf{x})}{\partial \mathbf{x}_j} = \begin{cases} -\dfrac{\mathbf{x}_j}{\|\mathbf{x}\|_1^2}, & j \neq i. \\[3mm] \dfrac{1}{\|\mathbf{x}\|_1} - \dfrac{\mathbf{x}_j}{\|\mathbf{x}\|_1^2}, & j = i. \end{cases} \tag{12}$$

Then,

$$\left\|\frac{\partial \delta_i(\mathbf{x})}{\partial \mathbf{x}}\right\|_2^2 = \sum_{j=1}^n \frac{\mathbf{x}_j^2}{\|\mathbf{x}\|_1^4} + \frac{1}{\|\mathbf{x}\|_1^2} - \frac{2\mathbf{x}_i}{\|\mathbf{x}\|_1^3} \leq \frac{\|\mathbf{x}\|_2^2}{\|\mathbf{x}\|_1^4} + \frac{1}{\|\mathbf{x}\|_1^2} = \frac{\|\mathbf{x}\|_2^2 + \|\mathbf{x}\|_1^2}{\|\mathbf{x}\|_1^4} \leq \frac{2}{\|\mathbf{x}\|_1^2} \leq 2. \quad (13)$$

Therefore, $\|\partial \delta_i(\mathbf{x})/\partial \mathbf{x}\|_2 \leq \sqrt{2} < 2$. Further, $\forall \mathbf{x}, \mathbf{x}' \in \mathbb{R}_+^n$, by mean value theorem, $\exists \alpha \in [0, 1]$ such that for $\mathbf{m} = \alpha \mathbf{x} + (1 - \alpha)\mathbf{x}'$, we have

$$\|\delta(\mathbf{x}) - \delta(\mathbf{x}')\|_1 = \sum_{i=1}^n |\langle \nabla_\mathbf{x} \delta_i(\mathbf{x})|_{\mathbf{x}=\mathbf{m}}, \mathbf{x} - \mathbf{x}'\rangle| \leq \sum_{i=1}^n \|\nabla_\mathbf{x} \delta_i(\mathbf{x})|_{\mathbf{x}=\mathbf{m}}\|_2 \|\mathbf{x} - \mathbf{x}'\|_2 \leq 2n\|\mathbf{x} - \mathbf{x}'\|_1.$$

Next we show that

$$\begin{aligned}
|\ell(\mathbf{a}, \mathbf{a}') - \ell(\mathbf{b}, \mathbf{b}')| &= |\ell(\mathbf{a}, \mathbf{a}') - \ell(\mathbf{b}, \mathbf{a}') + \ell(\mathbf{b}, \mathbf{a}') - \ell(\mathbf{b}, \mathbf{b}')| \\
&\leq |\ell(\mathbf{a}, \mathbf{b})| + |\ell(\mathbf{a}', \mathbf{b}')| \\
&= \|\delta(\mathbf{a}) - \delta(\mathbf{b})\|_1 + \|\delta(\mathbf{a}') - \delta(\mathbf{b}')\|_1 \\
&\leq 2n(\|\mathbf{a} - \mathbf{b}\|_1 + \|\mathbf{a}' - \mathbf{b}'\|_1).
\end{aligned} \quad (14)$$

$\square$

**Lemma D.** *Given two probability measures $\boldsymbol{\mu} \in \mathbb{R}_+^n$, $\boldsymbol{\nu} \in \mathbb{R}_+^n$ and a cost function $\mathbf{C}$ with $\mathbf{C}_{ij} \geq 0, \forall i, j$, the optimal transport cost function*

$$\mathcal{W}(\boldsymbol{\mu}, \boldsymbol{\nu}) = \min_{\gamma \in \Gamma} \langle \mathbf{C}, \gamma \rangle, \quad \Gamma = \{\gamma \in \mathbb{R}_+^{n \times n} : \gamma \mathbf{1} = \boldsymbol{\mu}, \gamma^T \mathbf{1} = \boldsymbol{\nu}\} \quad (15)$$

*is $\mathbf{C}_\infty$-Lipschitz w.r.t. $(\boldsymbol{\mu}, \boldsymbol{\nu})$ under the $L_1$ metric, i.e. $\forall(\boldsymbol{\mu}_1, \boldsymbol{\nu}_1)$ and $(\boldsymbol{\nu}_1, \boldsymbol{\nu}_2)$, we have*

$$|\mathcal{W}(\boldsymbol{\mu}_1, \boldsymbol{\nu}_1) - \mathcal{W}(\boldsymbol{\mu}_2, \boldsymbol{\nu}_2)| \leq \mathbf{C}_\infty \left(\|\boldsymbol{\mu}_1 - \boldsymbol{\mu}_2\|_1 + \|\boldsymbol{\nu}_1 - \boldsymbol{\nu}_2\|_1\right).$$

*Proof.* Consider the dual form of Eq. (15)

$$\mathcal{W}(\boldsymbol{\mu}, \boldsymbol{\nu}) = \max_{\boldsymbol{\alpha}, \boldsymbol{\beta} \in \mathbb{R}^n} \langle \boldsymbol{\alpha}, \boldsymbol{\mu} \rangle - \langle \boldsymbol{\beta}, \boldsymbol{\nu} \rangle, \quad \text{s.t. } \alpha_i - \beta_j \leq \mathbf{C}_{ij}, \forall i, j. \quad (16)$$

(Note that by replacing $\boldsymbol{\beta}$ in Eq. (2) in the main paper with $-\boldsymbol{\beta}$ we get Eq. (16) above). Denote the optimal solutions by $\boldsymbol{\alpha}^*$ and $\boldsymbol{\beta}^*$. Let $\alpha_{min}^* = \min(\boldsymbol{\alpha}^*)$ and $\alpha_{max}^* = \max(\boldsymbol{\alpha}^*)$ be the minimum and maximum values in $\boldsymbol{\alpha}^*$, respectively. Accordingly, let $\beta_{min}^* = \min(\boldsymbol{\beta}^*)$ and $\beta_{max}^* = \max(\boldsymbol{\beta}^*)$. First we show that $\alpha_{min}^* \geq \beta_{min}^*$ and $\alpha_{max}^* \geq \beta_{max}^*$. We prove $\alpha_{min}^* \geq \beta_{min}^*$ by contradiction. If $\alpha_{min}^* < \beta_{min}^*$, without loss of generality, let $\alpha_k^* = \alpha_{min}^*$. We can construct a new $\boldsymbol{\alpha}^{**}$ in which $\alpha_i^{**} = \alpha_i^*, i \neq k$ and $\alpha_k^{**} = \beta_{min}^*$. The new solution $(\boldsymbol{\alpha}^{**}, \boldsymbol{\beta}^*)$ does not violate the constraints in Eq. (16) and increases the objective. Therefore, $(\boldsymbol{\alpha}^*, \boldsymbol{\beta}^*)$ is not the optimal solution to Problem (16). A contradiction. $\alpha_{max}^* \geq \beta_{max}^*$ can be proved in a similar way (by decreasing $\beta_{max}^*$ to $\alpha_{max}^*$). Therefore, we have $\alpha_{min}^* \geq \beta_{min}^*$ and $\alpha_{max}^* \geq \beta_{max}^*$. Let $r$ denote the minimum value in $\boldsymbol{\alpha}^*$ and $\boldsymbol{\beta}^*$, i.e., $r = \min(\alpha_{min}^*, \beta_{min}^*)$. Let $\boldsymbol{\alpha}' = \boldsymbol{\alpha}^* - r$ and $\boldsymbol{\beta}' = \boldsymbol{\beta}^* - r$. So, $\min(\alpha_{min}', \beta_{min}') = 0$ and $(\boldsymbol{\alpha}', \boldsymbol{\beta}')$ is also the solution to Problem (16). In addition, we have $\alpha_{min}' \geq \beta_{min}'$ and $\alpha_{max}' \geq \beta_{max}'$ as well. As $\alpha_i' - \beta_j' \leq \mathbf{C}_{ij}, \forall i, j$, we have $\max(\alpha_{max}', \beta_{max}') \leq \mathbf{C}_\infty$. Therefore, $\forall \boldsymbol{\mu}$ and $\forall \boldsymbol{\nu}$, the solution values are in $[0, \mathbf{C}_\infty]$.

$$\left\|\begin{bmatrix} \dfrac{\partial \mathcal{W}(\boldsymbol{\mu}, \boldsymbol{\nu})}{\boldsymbol{\mu}} \\ \dfrac{\partial \mathcal{W}(\boldsymbol{\mu}, \boldsymbol{\nu})}{\boldsymbol{\nu}} \end{bmatrix}\right\|_\infty = \left\|\begin{bmatrix} \boldsymbol{\alpha}' \\ -\boldsymbol{\beta}' \end{bmatrix}\right\|_\infty \leq \mathbf{C}_\infty. \quad (17)$$

By applying Theorem $A$ with $p = \infty$ and $q = 1$, we know that $\mathcal{W}(\boldsymbol{\mu}, \boldsymbol{\nu})$ is $\mathbf{C}_\infty$-Lipschitz, i.e., $\forall(\boldsymbol{\mu}_1, \boldsymbol{\nu}_1)$ and $(\boldsymbol{\nu}_1, \boldsymbol{\nu}_2)$, we have $|\mathcal{W}(\boldsymbol{\mu}_1, \boldsymbol{\nu}_1) - \mathcal{W}(\boldsymbol{\mu}_2, \boldsymbol{\nu}_2)| \leq \mathbf{C}_\infty (\|\boldsymbol{\mu}_1 - \boldsymbol{\mu}_2\|_1 + \|\boldsymbol{\nu}_1 - \boldsymbol{\nu}_2\|_1)$.

$\square$

**Lemma E.** *Let $\mathcal{H}_i$ be a space of hypothesis functions, $i = 1, ..., n$. Each $h \in \mathcal{H}_i : \mathbb{R}^m \mapsto \mathbb{R}, i = 1, ..., n$. Let $\mathcal{F} = \mathcal{H}_1 \times \cdots \times \mathcal{H}_n$ be the space of mappings. Let $g : \mathbb{R}^n \mapsto \mathbb{R}$ be a L-Lipschitz*

*function under the $L_1$ metric. Let $S = \{I_1, ..., I_k\}$.*

*a) we have*

$$R_S(g \circ \mathcal{F}) \leq L \sum_{i=1}^n R_S(\mathcal{H}_i), \quad and \quad R_K(g \circ \mathcal{F}) \leq L \sum_{i=1}^n R_K(\mathcal{H}_i). \tag{18}$$

*b) Let*

$$\mathcal{F}' = \underbrace{\mathcal{H}_1 \times \cdots \times \mathcal{H}_n}_{n\ repetitions} \times \underbrace{\mathcal{I} \times \cdots \times \mathcal{I}}_{n\ repetitions}$$

*where $\mathcal{I}$ is a singleton function space with only the identity function which maps from $\mathbb{R}$ to $\mathbb{R}$, i.e., $\mathcal{I} = \{\phi\}$ and $\phi(x) = x$. We have*

$$R_S(g \circ \mathcal{F}') \leq L \sum_{i=1}^n R_S(\mathcal{H}_i), \quad and \quad R_K(g \circ \mathcal{F}') \leq L \sum_{i=1}^n R_K(\mathcal{H}_i). \tag{19}$$

*Proof.* a) Let $\boldsymbol{\sigma} = (\sigma_1, .., \sigma_K)^\intercal$, $\sigma_k \sim \{+1, -1\}$. Also, Let $\boldsymbol{\sigma}^i = (\sigma_1, .., \sigma_K)^\intercal, i = 1, ..., n$. Denote $h_1 \in \mathcal{H}_1, ..., h_n \in \mathcal{H}_n$ by $h \in \mathcal{H}$, and $h'_1 \in \mathcal{H}_1, ..., h'_n \in \mathcal{H}_n$ by $h' \in \mathcal{H}$.

$$
\begin{aligned}
K \cdot R_S(g \circ \mathcal{F}) &= \mathbb{E}_{\boldsymbol{\sigma}} \left[ \sup_{h \in \mathcal{H}} \sum_{k=1}^K \sigma_k \cdot g\left(h_1(I_k), ..., h_n(I_k)\right) \right] \\
&\leq \mathbb{E}_{\boldsymbol{\sigma}} \left[ \sup_{h \in \mathcal{H}} \sum_{k=1}^K \sigma_k L \left( \sum_{i=1}^n h_i(I_k) \right) \right] \\
&= L \sum_{i=1}^n \mathbb{E}_{\boldsymbol{\sigma}^i} \left[ \sup_{h_i \in \mathcal{H}_i} \sum_{k=1}^K \sigma_k^i h_i(I_k) \right] \\
&= K \cdot L \sum_{i=1}^n R_S(\mathcal{H}_i),
\end{aligned}
\tag{20}
$$

The first inequality in Eq. (20) is achieved because $g$ is $L$-Lipschitz, and for each $\sigma_k$ taking $+1$ and $-1$, we have

$$
\begin{aligned}
&\sup_{h \in \mathcal{H},\ h' \in \mathcal{H}} g(h_1(I_k), ..., h_n(I_k)) - g(h'_1(I_k), ..., h'_n(I_k)) \\
&= \sup_{h \in \mathcal{H},\ h' \in \mathcal{H}} |g(h_1(I_k), ..., h_n(I_k)) - g(h'_1(I_k), ..., h'_n(I_k))| \\
&\leq \sup_{h \in \mathcal{H},\ h' \in \mathcal{H}} L \sum_{i=1}^n |h_1(I_k) - h'_1(I_k)| + ... + |h_n(I_k) - h'_n(I_k)| \\
&= \sup_{h \in \mathcal{H},\ h' \in \mathcal{H}} L(h_1(I_k) + ... + h_n(I_k)) - L(h'_1(I_k) + ... + h'_n(I_k)) \\
&\leq \sup_{h \in \mathcal{H}} L(h_1(I_k) + ... + h_n(I_k)) + \sup_{h \in \mathcal{H}} -L(h_1(I_k) + ... + h_n(I_k)) \\
&= \sup_{h \in \mathcal{H}} L \sum_{i=1}^n h_i(I_k) + \sup_{h \in \mathcal{H}} -L \sum_{i=1}^n h_i(I_k)
\end{aligned}
\tag{21}
$$

The second equality in Eq. (20) is achieved because $h_i, i = 1, ..., n$ are independent of each other, and $\boldsymbol{\sigma}$ is broadcast to each $h_i$ as $\boldsymbol{\sigma}^i$.

By taking the expectation w.r.t. $S$ of both sides of $R_S(g \circ \mathcal{F}) \leq L \sum_{i=1}^n R_S(\mathcal{H}_i)$, we get $R_K(g \circ \mathcal{F}) \leq L \sum_{i=1}^n R_K(\mathcal{H}_i)$.

b) We only need to show that $R_S(\mathcal{I}) = 0$.

$$R_S(\mathcal{I}) = \frac{1}{K} \mathbb{E}_{\boldsymbol{\sigma}} \left[ \sup_{\phi \in \mathcal{I}} \sum_{k=1}^K \sigma_k \phi(x_k) \right] = \frac{1}{K} \mathbb{E}_{\boldsymbol{\sigma}} \left[ \sum_{k=1}^K \sigma_k x_k \right] = 0$$

$\square$

Now, we are ready to prove the two theorems we proposed in the main paper. Let

$$\mathcal{F}' = \underbrace{\mathcal{H} \times \cdots \times \mathcal{H}}_{n \text{ repetitions}} \times \underbrace{\mathcal{I} \times \cdots \times \mathcal{I}}_{n \text{ repetitions}} \tag{22}$$

and each $f' \in \mathcal{F}'$ can be considered as a mapping that maps from $(I, \mathbf{z})$ to $(\hat{\mathbf{z}}, \mathbf{z})$ or from $(I, \mathbf{t})$ to $(\hat{\mathbf{z}}, \mathbf{t})$. We restate Theorem 1 in the main paper below.

**Theorem 1.** *Assume that $\forall f \in \mathcal{F}$ and $(I, \mathbf{t}) \sim \tilde{\mathcal{D}}$, we have $\ell(\mathbf{t}, f(I)) \leq B$. Then, for any $0 < \delta < 1$, with probability of at least $1 - \delta$,*
*a) the upper bound of the generalization error is*

$$\mathcal{R}(\mathcal{D}, f_1^{\tilde{S}}, \ell_1) \leq \mathcal{R}(\tilde{\mathcal{D}}, f_1^{\tilde{\mathcal{D}}}, \ell_1) + 2nR_{\tilde{S}}(\mathcal{H}) + 5B\sqrt{2\log(8/\delta)/K} + \mathbb{E}_{(I,\mathbf{z})\sim\mathcal{D}}\|\mathbf{z} - \mathbf{t}\|_1,$$

*b) the lower bound of the generalization error is*

$$\mathcal{R}(\mathcal{D}, f_1^{\tilde{S}}, \ell_1) \geq \left| \mathbb{E}_{(I,\mathbf{z})\sim\mathcal{D}}\|\mathbf{z} - \mathbf{t}\|_1 - \mathcal{R}(\tilde{\mathcal{D}}, f_1^{\tilde{S}}, \ell_1) \right|.$$

*Proof.* Let $\hat{\mathbf{z}} = f_1^{\tilde{S}}(I)$. Let each $\mathbf{t}$ be the Gaussian smoothed ground truth of the corresponding real ground truth $\mathbf{z}$. The idea to prove a) and b) is to build the relationship among the real ground truth $\mathbf{z}$, the Gaussian smoothed ground truth $\mathbf{t}$, and the predicted density map $\hat{\mathbf{z}}$ by using triangle inequalities, i.e., $\|\mathbf{z} - \hat{\mathbf{z}}\|_1 \leq \|\mathbf{z} - \mathbf{t}\|_1 + \|\hat{\mathbf{z}} - \mathbf{t}\|_1$ and $\|\mathbf{z} - \hat{\mathbf{z}}\|_1 \geq |\|\mathbf{z} - \mathbf{t}\|_1 - \|\hat{\mathbf{z}} - \mathbf{t}\|_1|$.
a)

$$
\begin{aligned}
\mathcal{R}(\mathcal{D}, f_1^{\tilde{S}}, \ell_1) &= \mathbb{E}_{(I,\mathbf{z})\sim\mathcal{D}}[\|\mathbf{z} - f_1^{\tilde{S}}(I)\|_1] \\
&= \mathbb{E}_{(I,\mathbf{z})\sim\mathcal{D}}[\|\mathbf{z} - \hat{\mathbf{z}}\|_1] \\
&\leq \mathbb{E}_{(I,\mathbf{z})\sim\mathcal{D}}\left[\|\mathbf{z} - \mathbf{t}\|_1 + \|\hat{\mathbf{z}} - \mathbf{t}\|_1\right] \\
&= \mathcal{R}(\tilde{\mathcal{D}}, f_1^{\tilde{S}}, \ell_1) + \mathbb{E}_{(I,\mathbf{z})\sim\mathcal{D}}\|\mathbf{z} - \mathbf{t}\|_1,
\end{aligned} \tag{23}
$$

where the inequality follows by the triangle inequality. Further, we have

$$
\begin{aligned}
\mathcal{R}(\tilde{\mathcal{D}}, f_1^{\tilde{S}}, \ell_1) &\leq \mathcal{R}(\tilde{\mathcal{D}}, f_1^{\tilde{\mathcal{D}}}, \ell_1) + 2R_{\tilde{S}}(\ell_1 \circ \mathcal{F}') + 5B\sqrt{2\log(8/\delta)/K} \\
&\leq \mathcal{R}(\tilde{\mathcal{D}}, f_1^{\tilde{\mathcal{D}}}, \ell_1) + 2nR_{\tilde{S}}(\mathcal{H}) + 5B\sqrt{2\log(8/\delta)/K},
\end{aligned} \tag{24}
$$

where the first inequality follows by applying Theorem B, and the second inequality follows by combining Lemmas A and E (b). Finally, a) is proved by combining Eqs. (23) and (24).
b)

$$
\begin{aligned}
\mathcal{R}(\mathcal{D}, f_1^{\tilde{S}}, \ell_1) &= \mathbb{E}_{(I,\mathbf{z})\sim\mathcal{D}}[\|\mathbf{z} - f_1^{\tilde{S}}(I)\|_1] \\
&= \mathbb{E}_{(I,\mathbf{z})\sim\mathcal{D}}[\|\mathbf{z} - \hat{\mathbf{z}}\|_1] \\
&\geq \mathbb{E}_{(I,\mathbf{z})\sim\mathcal{D}}\left[|\|\mathbf{z} - \mathbf{t}\|_1 - \|\hat{\mathbf{z}} - \mathbf{t}\|_1|\right] \\
&= \mathbb{E}_{(I,\mathbf{z})\sim\mathcal{D}}[\max(\|\mathbf{z} - \mathbf{t}\|_1 - \|\hat{\mathbf{z}} - \mathbf{t}\|_1, \|\hat{\mathbf{z}} - \mathbf{t}\|_1 - \|\mathbf{z} - \mathbf{t}\|_1)] \\
&\geq \max\left(\mathbb{E}_{(I,\mathbf{z})\sim\mathcal{D}}[\|\mathbf{z} - \mathbf{t}\|_1 - \|\hat{\mathbf{z}} - \mathbf{t}\|_1], \mathbb{E}_{(I,\mathbf{z})\sim\mathcal{D}}[\|\hat{\mathbf{z}} - \mathbf{t}\|_1 - \|\mathbf{z} - \mathbf{t}\|_1]\right) \\
&= \max\left(\mathbb{E}_{(I,\mathbf{z})\sim\mathcal{D}}[\|\mathbf{z} - \mathbf{t}\|_1] - \mathcal{R}(\tilde{\mathcal{D}}, f_1^{\tilde{S}}, \ell_1), \mathcal{R}(\tilde{\mathcal{D}}, f_1^{\tilde{S}}, \ell_1) - \mathbb{E}_{(I,\mathbf{z})\sim\mathcal{D}}[\|\mathbf{z} - \mathbf{t}\|_1]\right) \\
&= \left| \mathbb{E}_{(I,\mathbf{z})\sim\mathcal{D}}\|\mathbf{z} - \mathbf{t}\|_1 - \mathcal{R}(\tilde{\mathcal{D}}, f_1^{\tilde{S}}, \ell_1) \right|.
\end{aligned} \tag{25}
$$

The first inequality uses the triangle inequality, and the second inequality uses Jensen's inequality.
$\square$

We restate Theorem 2 in the main paper below.

**Theorem 2.** *Assume that $\forall f \in \mathcal{F}$ and $(I, \mathbf{z}) \sim \mathcal{D}$, we have $\|\mathbf{z}\|_1 \geq 1$, $\|f(I)\|_1 \geq 1$ (can be satisfied by adding a dummy dimension with value of 1 to both $\mathbf{z}$ and $f(I)$) and $\ell_C(\mathbf{z}, f(I)) \leq B$. Then, for any $0 < \delta < 1$, with probability of at least $1 - \delta$*
*a) the generalization error bound of the counting loss is*

$$\mathcal{R}(\mathcal{D}, f_C^S, \ell_C) \leq \mathcal{R}(\mathcal{D}, f_C^{\mathcal{D}}, \ell_C) + 2nR_S(\mathcal{H}) + 5B\sqrt{2\log(8/\delta)/K}$$

*b) the generalization error bound of the OT loss is*

$$\mathcal{R}(\mathcal{D}, f_{OT}^S, \ell_{OT}) \leq \mathcal{R}(\mathcal{D}, f_{OT}^{\mathcal{D}}, \ell_{OT}) + 4\mathbf{C}_\infty n^2 R_S(\mathcal{H}) + 5\mathbf{C}_\infty \sqrt{2\log{(8/\delta)}/K}$$

*c) the generalization error bound of the TV loss is*

$$\mathcal{R}(\mathcal{D}, f_{TV}^S, \ell_{TV}) \leq \mathcal{R}(\mathcal{D}, f_{TV}^{\mathcal{D}}, \ell_{TV}) + n^2 R_S(\mathcal{H}) + 5\sqrt{2\log{(8/\delta)}/K}$$

*d) the generalization error bound of the overall loss is*

$$\mathcal{R}(\mathcal{D}, f^S, \ell) \leq \mathcal{R}(\mathcal{D}, f^{\mathcal{D}}, \ell) + (2n + 4\lambda_1\mathbf{C}_\infty n^2 + \lambda_2 N n^2)R_S(\mathcal{H})$$
$$+5(B + \lambda_1\mathbf{C}_\infty + \lambda_2 N)\sqrt{2\log{(8/\delta)}/K}$$

*where $\mathbf{C}_\infty$ is the maximum cost in the cost matrix in OT, and $N = \sup\{\|\mathbf{z}\|_1 \mid \forall(I, \mathbf{z}) \sim \mathcal{D}\}$ is the maximum count number over a dataset.*

*Proof.* We prove a), b) and c) by observing that the original hypothesis space can be decomposed into the space of feature (density map) extraction functions and a loss function.

a)
$$\mathcal{R}(\mathcal{D}, f_C^S, \ell_C) \leq \mathcal{R}(\mathcal{D}, f_C^{\mathcal{D}}, \ell_C) + 2R_S(\ell_C \circ \mathcal{F}') + 5B\sqrt{2\log{(8/\delta)}/K}$$
$$\leq \mathcal{R}(\mathcal{D}, f_C^{\mathcal{D}}, \ell_C) + 2nR_S(\mathcal{H}) + 5B\sqrt{2\log{(8/\delta)}/K},$$

where the first inequality follows by applying Theorem B, and the second inequality follows by combining Lemmas B and E (b).

b)
$$\mathcal{R}(\mathcal{D}, f_{OT}^S, \ell_{OT}) \leq \mathcal{R}(\mathcal{D}, f_{OT}^{\mathcal{D}}, \ell_{OT}) + 2R_S(\ell_{OT} \circ \mathcal{F}') + 5\mathbf{C}_\infty\sqrt{2\log{(8/\delta)}/K}$$
$$\leq \mathcal{R}(\mathcal{D}, f_{OT}^{\mathcal{D}}, \ell_{OT}) + 4\mathbf{C}_\infty n^2 R_S(\mathcal{H}) + 5\mathbf{C}_\infty\sqrt{2\log{(8/\delta)}/K},$$

where the first inequality follows by applying Theorem B, and the objective value of OT loss is bounded by $\mathbf{C}_\infty$, since

$$\mathcal{W}(\boldsymbol{\mu}, \boldsymbol{\nu}) = \sum_{ij}\mathbf{C}_{ij}\gamma_{ij} \leq \sum_{ij}\mathbf{C}_\infty\gamma_{ij} = \mathbf{C}_\infty,$$

and the second inequality follows by combining Lemmas C, D and E (b).

c)
$$\mathcal{R}(\mathcal{D}, f_{TV}^S, \ell_{TV}) \leq \mathcal{R}(\mathcal{D}, f_{TV}^{\mathcal{D}}, \ell_{TV}) + 2R_S(\ell_{TV} \circ \mathcal{F}') + 5\sqrt{2\log{(8/\delta)}/K}$$
$$\leq \mathcal{R}(\mathcal{D}, f_{TV}^{\mathcal{D}}, \ell_{TV}) + n^2 R_S(\mathcal{H}) + 5\sqrt{2\log{(8/\delta)}/K},$$

where the first inequality follows by applying Theorem B, and the objective value of the TV loss is bounded by 1, since

$$\frac{1}{2}\left\|\frac{\mathbf{x}}{\|\mathbf{x}\|_1} - \frac{\mathbf{x}'}{\|\mathbf{x}'\|_1}\right\|_1 \leq \frac{1}{2}\left(\left\|\frac{\mathbf{x}}{\|\mathbf{x}\|_1}\right\|_1 + \left\|\frac{\mathbf{x}'}{\|\mathbf{x}'\|_1}\right\|_1\right) = 1,$$

and the second inequality follows by combining Lemmas C and E (b).

d) Recall that the overall loss is defined as

$$\ell(\mathbf{z}, \hat{\mathbf{z}}) = \ell_C(\mathbf{z}, \hat{\mathbf{z}}) + \lambda_1\ell_{OT}(\mathbf{z}, \hat{\mathbf{z}}) + \lambda_2\|\mathbf{z}\|_1\ell_{TV}(\mathbf{z}, \hat{\mathbf{z}}).$$

By combining a), b), and c) we obtain d) immediately. □

## 2 Experimental Details

### 2.1 Datasets

We perform experiments on four challenging crowd counting datasets: UCF-QNRF, NWPU, ShanghaiTech, and UCF-CC-50. The **UCF-QNRF** dataset [5] contains 1201 training and 334 test images of variable sizes, with 1.25 million dot annotations. The number of people in each image varies from 49 to 12,865 with the average being 815. The **ShanghaiTech** [13] dataset consists of two parts. Part A contains 482 images collected from the web, and Part B contains 716 images collected on the streets of Shanghai. Images in Part A contains more people than images in Part B; the average

counts for Part A and Part B are 501 and 124, respectively. Part A has 300 training and 182 test images, while Part B has 400 training and 316 test images. The **UCF-CC-50** dataset [4] contains 50 images from the web, with the minimum, average, and maximum numbers of people being 94, 1280, and 4532. Given the small size of the dataset, we perform five-fold cross validation and report the average result, as also done in previous works [12, 13, 1, 8, 10]. The **NWPU** [11] dataset is the latest and largest crowd counting dataset comprising of 5,109 images taken from the web and video sequences. The dataset consists of more than 2.1 million annotated people, and the count varies between 0 and 20,033. This dataset covers a wider range of lighting conditions, image counts, and crowd appearance. The dataset is split into 3,109 training, 500 validation, and 1500 test images. The ground truth counts for test images are not released, and the results on the test set must be obtained by submitting to the evaluation server `https://www.crowdbenchmark.com/nwpucrowd.html`. We report performance on both the validation and test sets.

## 2.2 Evaluation Metrics

Following previous works [8, 5, 3, 4, 13], we use the following metrics: Mean Absolute Error (MAE), Root Mean Squared Error (RMSE), and mean Normalized Absolute Error (NAE). Given $K$ test images, and let $C_k$ and $\hat{C}_k$ be the ground truth and predicted counts for the $k^{th}$ image.

$$MAE = \frac{1}{K}\sum_{k=1}^{K}|C_k - \hat{C}_k|; RMSE = \sqrt{\frac{1}{K}\sum_{k=1}^{K}(C_k - \hat{C}_k)^2}; NAE = \frac{1}{K}\sum_{k=1}^{K}\frac{|C_k - \hat{C}_k|}{C_k}.$$

## 2.3 Implementation Details

For a fair comparison, we use the same network as in Bayesian loss [6]. The network consists of standard feature extraction backbone, followed by a density map estimation head. VGG-19 is used as the backbone, with the removal of the last pooling layer and the subsequent fully connected layers. The output of the backbone is upsampled to 1/8 of the input image size by bilinear interpolation. The density map prediction head has two $3\times3$ convolutional layers with 256 and 128 channels, followed by one $1\times1$ convolutional layer.

Due to large sizes of images in UCF-QNRF and NWPU, following [6, 11, 5] we limit the shorter size of image within 2048 and 1920 for UCF-QNRF and NWPU respectively. Random crops are taken for training. We use the same crop sizes as previous works [6, 11]: 256 for ShanghaiTech Part A and UCF-CC-50, 512 for ShanghaiTech Part B and UCF-QNRF, and 384 for NWPU.

## 3 Visualization

We analyze the counting error distribution of DM-Count on the UCF-QNRF dataset. Fig. 1a shows the overall counting error distribution. Most images have small counting errors, and the median counting error is 42. Fig. 1b visualizes the correlation between the counting error and the ground truth total count. A majority of the images have small counting errors. As the ground truth count increases, counting error tends to increase as well.

Fig. 2, 3, 4, 5 display more examples of the predicted density maps of the Pixel-wise loss, Bayesian loss, and our proposed DM-Count. In all these figures, DM-Count produces better count numbers, and sharper density maps than the other two methods.

Fig. 6 and Fig. 7 shows the predicted density maps of DM-Count on the NWPU validation set and test set, respectively. From these two figures we can see that DM-Count does well in producing density maps and locating people in various complicated crowded scenes.

To investigate in which cases DM-Count fails to perform well, we display four images for which DM-Count produces the highest MAE on the UCF-QNRF dataset and NWPU validataion dataset in Fig. 8. As we can observe from this figure, these images either have extremely dense crowds (row 1, 3, 4) or have poor illumination (row 2). The crowd is hard to count even for human observers because of extremely small people and dense crowds or poor lighting conditions.

Figure 1: **Counting error analysis.** 1a shows the counting error distribution of DM-Count on the UCF-QNRF dataset. 1b shows the correlation between counting error and ground truth total count.

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

**(a) Image.** Count: 564

**(b) Pixel-wise loss.** Count: 604.6, PSNR: 29, SSIM: 0.04

**(c) Bayesian loss.** Count: 592.8, PSNR: 28, SSIM: 0.05

**(d) DM-Count (proposed).** Count: 569.8, PSNR: 32, SSIM: 0.08

Figure 2: **Density map visualization on the UCF-QNRF dataset.** Best viewed in color.

(a) **Image.** Count: 674

(b) **Pixel-wise loss.** Count: 825.5, PSNR: 37, SSIM: 0.68

(c) **Bayesian loss.** Count: 809.3, PSNR: 37, SSIM: 0.66

(d) **DM-Count (proposed).** Count: 769, PSNR: 41, SSIM: 0.75

Figure 3: **Density map visualization on the UCF-QNRF dataset.** Best viewed in color.

**(a) Image.** Count: 2530

**(b) Pixel-wise loss.** Count: 1786.9, PSNR: 33, SSIM: 0.23

**(c) Bayesian loss.** Count: 1948.1, PSNR: 30, SSIM: 0.14

**(d) DM-Count (proposed).** Count: 2328, PSNR: 32, SSIM: 0.21

Figure 4: **Density map visualization on the UCF-QNRF dataset.** Best viewed in color.

(a) **Image.** Count:2494

(b) **Pixel-wise loss.** Count: 1926.8, PSNR: 32, SSIM: 0.45

(c) **Bayesian loss.** Count: 2034.1, PSNR: 32, SSIM: 0.50

(d) **DM-Count (proposed).** Count: 2057.9, PSNR: 35, SSIM: 0.53

Figure 5: **Density map visualization on the UCF-QNRF dataset.** Best viewed in color.

| Image | DM-Count Predicted Result |
|-------|---------------------------|
|  |  |
| Count: 1748 | Predicted Count: 1634.4 |
|  |  |
| Count: 3470 | Predicted Count: 3116.1 |
|  |  |
| Count: 1270 | Predicted Count: 1335.3 |
|  |  |
| Count: 239 | Predicted Count: 247.3 |

Figure 6: **Density map visualization on the NWPU validation set.** Best viewed in color.

| Image | DM-Count Predicted Result |
|:---:|:---:|
|  |  |
| Count: unavailable to the public | Predicted Count: 3879.2 |
|  |  |
| Count: unavailable to the public | Predicted Count: 844.7 |
|  |  |
| Count: unavailable to the public | Predicted Count: 1279.3 |
|  |  |
| Count: unavailable to the public | Predicted Count: 901.5 |

Figure 7: **Density map visualization on the NWPU test set.** Note that the ground truths of NWPU's test set are unavailable to the public.

| Image | DM-Count Predicted result |
|:---:|:---:|

Count: 2370 — Predicted count: 1401

Count: 1443 — Predicted count: 678.6

Count: 12924 — Predicted count: 6524.9

Count: 7122 — Predicted count: 4860

Figure 8: **Failure case visualization**. Images with highest counting errors by DM-Count on the UCF-QNRF dataset (top two rows) and NWPU validation dataset (bottom two rows).