[Reviews · NeurIPS 2020]

Review 1

Summary and Contributions: The paper presents a distribution matching based loss for crowd counting. Most existing methods use Gaussians at head locations to generate heat map which is then used as targets for training the networks. The authors argue that this method could hurt the performance, and hence propose a optimal transport based distribution matching loss to train the network. The authors show that the using this loss results in much tighter bound on the error. The authors evaluate their methods on 4 datasets and show improvements on some of them.

Strengths: 1. Representing targets for crowd counting is a known issue. There is a recent interest in addressing this. The proposed method is an interesting approach for this issue. 2. The authors show theoretically and empirically that the proposed solution results in lower error.

Weaknesses: 1. Novelty - The contributions of the paper in terms of novelty are: (i) The idea of using OT based distribution matching loss, (ii) theoretical results showing that such loss results in lower error, (iii) empirical results showing that such a loss indeed results in lower error. For the crowd counting community, this may be considered as considerable contributions. However, the paper does not address if this is of interest to the broader vision/ml community - which is expected for a Neurips kind of venue. For example, the authors could have considered a broader set of applications like object detection for evaluating their method. 2. Related work - The authors have not considered several recent works like [1-8] in their related works discussion. Further, the authors should have given a better background for the recent works that have focussed on improving representation of ground-truth [2,9,10] for training the networks. 3. Comparison of results: The authors claim that they obtain significant improvements in 4 datasets compared to recent SOTA. First - they have conveniently left out comparisons to recent methods like [2,11,12,13,14] that have better/comparable results compared to the proposed method. Second, their improvements over SOTA cannot be considered significant in all the datasets. In many cases, the numbers are too close. For example: (i) UCF-QNRF, their MAE is 85.6 vs BL which is 88.7 (I agree MSE is much better), (ii) ShanghaiTech numbers are too close to BL. (iii) UCF-CC-50, the difference in MAE is only 1.2 (as compared to CAN) and the MSE for OT is much worse compared to CAN. [1] From Open Set to Closed Set: Counting Objects by Spatial Divide-and-Conquer. ICCV 19 [2] Multi-Level Bottom-Top and Top-Bottom Feature Fusion for Crowd Counting. ICCV 19 [3] Leveraging unlabeled data for crowd counting by learning to rank. [4] Top-down feed- back for crowd counting convolutional neural network. AAAI 2019 [5] Where are the Blobs: Counting by Localization with Point Supervision. ECCV 2018 [6] Crowd Counting via Adversarial Cross-Scale Consistency Pursuit. CVPR 2018 [7] DecideNet: Counting Varying Density Crowds Through Attention Guided Detection and Density. CVPR 2018 [8] Crowd Counting with Deep Negative Correlation Learning [9] Adaptive Density Map Generation for Crowd Counting. ICCV 19 [10] Improving Dense Crowd Counting Convolutional Neural Networks using Inverse k-Nearest Neighbor Maps and Multiscale Upsampling. VISAPP 20 [11] Crowd Counting with Deep Structured Scale Integration Network. ICCV 19 [12] Relational Attention Network for Crowd Counting. ICCV 19 [13] PaDNet: Pan-Density Crowd Counting. TIP 19. [14] Ha-ccn: Hi- erarchical attention-based crowd counting network. TIP 2019

Correctness: Yes

Clarity: Yes

Relation to Prior Work: No.

Reproducibility: Yes

Additional Feedback: Update after the rebuttal --------------------------------- I had 3 main issues with this paper - 1. impact on the broader vision community, 2. lack of discussion on related work, and 3. missing comparisons. 1. Impact to the broader vision community - I had asked for the authors to discuss about the broader impact which is important for Neurips. Similar concerns were raised by R2, where R2 asked if it is possible to “transfer DM-count similar tasks like key point regression”. Instead of focusing on this in their rebuttal, the authors go on to argue how crowd counting is very appropriate for Neurips by stating that “4 dozen papers are published at top tier conferences”. This statement does not address if their solution is applicable to other problems like what R2 suggested. For now, the paper is narrowly focussed on crowd counting. 2. Related work: The authors state that it is not possible to cite each and every paper. While this is true - they are expected to cite the papers which have similar motivation of improving the representation of density maps [2,9,10]. These works also focus on improving the density maps for better learning - however, they are not being discussed. They refer to lines 82-84 in their rebuttal - but I don’t see these references. 3. Missing comparisons - My main concern with this is that they missed out comparing with methods which were better than them, which is kind of misleading to the reader. I do understand, it may not be possible to outperform all the methods in all the datasets - and that is okay. However, a discussion why this Is the case would be helpful for the reader. Additionally, some of their statements like (i) “Our method ranked first in the leaderboard 26 at the time of submission, reducing SOTA error from 105 to 88” is misleading - since BL has achieved 88.7 and it is an ICCV 19 work, (ii) Also, their method in the table is highlighted for ShanghaiTech-B, but [11] performs better. After carefully reading the rebuttal, the authors have not completely addressed my concerns, especially with respect to the impact on the broader set of vision problems. However, as I had stated in my original comments, I do believe that the contributions are novel especially to the crowd counting community. Also, after reading comments from other reviewers and the rebuttal for them, I reconsider my earlier decision and upgrade the rating of the paper. The authors are recommended to include a discussion on how their method could be applied to other similar problems, and also discuss at least the relevant related work if not all of them.


Review 2

Summary and Contributions: This work demonstrates that using the Gaussian kernel to smooth the ground truth dot annotations can hurt the generalisation bound and transfers the crowd counting task to a distribution matching problem by using the proposed DM-Count. There are three loss items in DM-Count: the counting loss, the OT loss, and the Total Variation (TV) loss. The performance of DM-Count on the widely used datasets is better than state-of-the-art results.

Strengths: It is a good idea to use the Optimal Transport on the Crowd Counting and a detailed theoretical proof is provided in this paper. The experiments demonstrate the effectiveness of the proposed DM-Count. This paper is well written and meaningful to the community.

Weaknesses: 1. In #141, “The OT loss will approximate well the dense areas of the crowd, but the approximation might be poorer for the low density areas of the crowd”. I wonder why there is a performance gap in the crowd areas and low-density areas for the OT loss. More details should be provided for explanation. 2. In #248, “In all experiments, DM-Count outperforms all other methods except CAN under MSE in NWPU (where they are comparable)”. Why the DM-Count performs bad on RMSE compared with CAN? 3. In Table 3, without using the TV loss, the performance of DM-Count is worse than Bayesian Loss (in Table 2). I think the OT loss may not robust. 4. Is this possible to transfer the DM-Count to other similar tasks, like keypoint regression? I think it is interesting if the DM-Count could solve the crowded problems in the human-pose estimation.

Correctness: Yes.

Clarity: Yes.

Relation to Prior Work: Yes.

Reproducibility: Yes

Additional Feedback: Please address the problems mentioned in “Weaknesses”. Besides, the “Ablation Studies” mentioned in the supplementary is important for readers to learn more details about each component and I suggest appending it to the paper if possible.


Review 3

Summary and Contributions: This paper proposes a new objective function for crowd counting, which minimizes the Wasserstein distance between the predicted normalized density map and the ground-truth point maps without using the Gaussian assumption. Generalization bounds and theoretical analysis are also provided in this paper.

Strengths: + This paper is well written and easy to understand. + This paper has a good motivation and a solid theoretical grounding. + Although the Wasserstein distance has been widely applied in GAN and other applications. This is the first time it is applied in crowd counting tasks.

Weaknesses: 1 There are three terms in the loss function, the effects of the introduced hyper-parameters should be studied. 2 The third loss is named the total variation loss. However, the "total variation" is a terminology that should be clearly defined and explained. The derivation of Eq. 6 should also be further explained. 3 The two-dimention Wasserstein distance doesn't have a close form solution, this paper applies the Sinkhorn algorithm to get the approximate solution. As mentioned in this paper, iterations are needed before each gradient descend. What is the maximum iterations you set in the experiment? Will these iterations significantly slow down the training speed?

Correctness: The defination of the terminology is unclear.

Clarity: This paper is well written.

Relation to Prior Work: The relation to prior work has been clearly discussed.

Reproducibility: No

Additional Feedback: If my concerns are addressed, I am willing to increase the score.


Review 4

Summary and Contributions: This paper proposes a Distribution Matching method for crowd counting. It uses optimal transport to measure the similarity between the normalized predicted/GT density maps.

Strengths: + Show that imposing Gaussians to annotations will hurt the generalization performance of counting. + DM-Count may be a new direction for counting. It gets rid of Gaussian-smoothing density map. + Propose OT loss and achieve the SOTA.

Weaknesses: - Ablation study is not adequate. The author should add more analysis. For example, the results of single loss should be shown (only OT loss, only TV loss). - In addition to visualization results, the author should evaluate the quantitative counting performance in high-density region to show the improvements. - Minor issues: missing the reference of PCC-Net-VGG in Table 2.

Correctness: Yes.

Clarity: Good.

Relation to Prior Work: Yes. The paper shows the differences with other works, such as Bayesian loss.

Reproducibility: Yes

Additional Feedback: If the author can release the code, I think it is very useful for the community. After the rebuttal: Although the authors respond to my issues well, I think this paper has some key weakness after reading other reviews and feedbacks, especially R1. So my final score is “6. Marginally above the acceptance threshold”.

[Author Response · NeurIPS 2020]

**Reviewer 1.** **Q1:** *The paper does not address if this is of interest to broader vision/ml community.* **A1:** Crowd counting is a well established research problem in Computer Vision, with more than four dozen papers published at top-tier conferences in the last two years, including the 12 papers mentioned by R1. Our paper is motivated by crowd counting applications, but it actually addresses the underlying fundamental problem: spatial density estimation, which has broader interest to machine learning and other communities. We believe that crowd counting is a very appropriate and salient evaluation domain for spatial density estimation, especially in the age of social distancing. In addition, almost all papers including those listed by R1 used the Gaussian smoothed Ground Truth (GGT), so we believe that a theoretical analysis of GGT is necessary and that NeurIPS is a proper venue to present our theoretical results.

**Q2:** *Missing discussions of several recent works, especially [2,9,10] which focused on improving representation of ground-truth for training the networks.* **A2:** We would be happy to cite these papers in our revised paper. Notably, it is neither possible nor necessary to cite every single crowd counting paper given the large number of papers. We could only discuss the major approaches and representative papers. All the listed papers except [5] used the GGT, which is discussed in lines 38–41. Refs [2,9,10] belong to the group of papers that use adaptive kernel widths, discussed in lines 82-84. Ref [5] belongs to the detection-then-count approach, line 73–76.

**Q3:** *Missing comparisons to recent methods that have better/comparable results. The improvements are insignificant.*

**A3:** Thanks for the references, and this table compares with methods in the references. Considering all four datasets as a whole, our method outperforms the other methods. Considering individual datasets, QNRF is the most difficult and largest one, and the performance gaps between our method and others are wide. For ShTech datasets, our method performed comparable to the bests. On UCF-CC 50, the comparison should be taken with a grain of salt due to small number of images (50) and different ways of data splitting for five-fold cross validation. Moreover, none of these methods were evaluated on the large-scale NWPU dataset. Our method ranked first in the leaderboard at the time of submission, reducing SOTA error from 105 to 88.

| | QNRF | | ShTech A | | ShTech B | | CC-50 | |
|---|---|---|---|---|---|---|---|---|
| | MAE | RMSE | MAE | RMSE | MAE | RMSE | MAE | RMSE |
| [2] | 97.5 | 165.2 | 60.2 | **94.1** | 8.0 | 15.5 | 233 | 300 |
| [11] | 99.1 | 159.2 | 60.6 | 96 | **6.8** | **10.3** | 216 | 302 |
| [12] | 111 | 190 | 59.4 | 102 | 7.9 | 12.9 | 239 | 319 |
| [13] | 96.5 | 170.2 | **59.2** | 98.1 | 8.1 | 12.2 | **185** | 278 |
| [14] | 118 | 180.4 | 62.9 | **94.9** | 8.1 | 13.4 | 256 | 348 |
| CAN | 107 | 183 | 62.3 | 100.0 | 7.8 | 12.2 | 212 | **243** |
| BL | 88.7 | 154.8 | 62.8 | 101.8 | 7.7 | 12.7 | 229 | 308 |
| Ours | **85.6** | **148.3** | 59.7 | 95.7 | **7.4** | **11.8** | 211 | 291 |

**Reviewer 2.** **Q4:** *DM-Count is worse than CAN in RMSE on UCF-CC-50 and NWPU.* **A4:** One reason is that CAN is trained by minimizing the MSE loss, while DM-Count optimizes for the MAE loss. On NWPU, the RMSEs are close (388.6 vs 386.5). The comparison on UCF-CC-50 should be taken with a grain of salt, as explained in Answer A3.

**Q5:** *Why OT loss approximate well for dense regions, but poorer for the sparse regions.* **A5:** Dense areas have higher probability values than sparse areas, and in general more probability masses must be transported between dense areas. The OT loss is therefore dominated by loss from dense areas. The dense areas have higher priority in the optimization, especially when using a finite number of Sinkhorn iterations for computing the OT distance.

**Q6:** *Without using the TV loss, DM-Count performed worse than Bayesian Loss (BL).* **A6:** This is due to the pre-mature stopping of the Sinkhorn algorithm. Tab. 3 reports the performance with 100 Sinkhorn iterations. Using 200 Sinkhorn iterations, OT loss + Counting loss outperforms BL with MAE 85 and MSE 154. As the TV loss treats both dense and sparse areas equally, sparse areas can be optimized well with TV loss and fewer Sinkhorn iterations (100).

**Q7:** *Is this possible to transfer the DM-Count to other similar tasks, like keypoint regression?* **A7:** Thanks for this interesting suggestion. We think it may be possible, given that the GT for keypoint estimation is also a dot map.

**Reviewer 3.** **Q8:** *The effects of the hyper-parameters should be studied.* **A8:** On QNRF, by fixing $\lambda_2$ to 0.01 and tuning $\lambda_1$ from 0.01, 0.05 to 0.1, the MAE varies from 87.2, 86.2 to 85.6. By fixing $\lambda_1$ to 0.1 and changing $\lambda_2$ from 0.01, 0.05 to 0.1, the MAE varies from 85.6, 87.8 to 88.5. See also A5, A6 for the effects of the OT loss and TV loss.

**Q9:** *"Total variation should be clearly defined and explained. The derivation of Eq. 6 should be further explained."*

**A9:** We will clarify that in the context of training loss, Total Variation refers to the total variation distance of two probability measures, not the total variation of a function. A formal definition can be found in Definition 2.4, pg 83, Tsybakov: Introduction to Nonparametric Estimation. Eq. 6 in our paper is Lemma 2.1, pg 84. We will cite and clarify.

**Q10:** *What is the maximum Sinkhorn iterations in the experiments? Will these iterations significantly slow down the training speed?* **A10:** We experiment with different numbers of iterations in Tab 1 of the supplementary material. The performance plateaus after 100 iterations. In our experiments, we used a maximum of 100 iterations, and the OT part takes 30% of the total training time. Thus the OT part does not significantly slow down the training speed.

**Q11:** *Reproducibility* **A11:** Implementation details are provided in Sec 2.3 of the supplementary material.

**Reviewer 4.** **Q12:** *Missing ablation studies of using single loss: only OT loss, only TV loss.* **A12:** the OT and TV losses take normalized density maps (probability distributions) as inputs (Eqs. 4 and 6), so it is not possible to obtain the absolute count using either loss by itself. The counting loss is always needed. In Tab. 3, we showed the performance of counting loss + OT loss (or TV loss). Besides, we will report the hyper-parameter study (see Answers A8).

**Q13:** *Quantitative performance in high-density region.* **A13:** The MAEs in high density images (Ground truth count over 1000 in QNRF test set) are 211 (DM-Count), 238 (BL) and 311 (pixel-wise loss). DM-Count outperforms two baselines significantly for high density images. We will address other minor issues and release code upon acceptance.

[Meta-Review · NeurIPS 2020]

The initial ratings were 3577. The main concerns were: 1) contribution focused on a specific application of crowd counting; 2) missing comparisons; 3) missing ablation studies on hyperparameter selection. In the response, (1) authors argue they address a fundamental problem in spatial density estimation, which has broad impact, and they use crowd counting, which is is a well established CV task, for evaluation; (2) provide a table of comparisons showing improved results on large-scale datasets, NWPU and QNRF; (3) provide the ablation study results. After the response and discussion, R1 upgraded from 3 to 6, while R4 downgraded 7 to 6. R3's concerns were addressed and also upgraded to 6. The final ratings were 6667. After reading the reviews and responses, the AC agrees with the authors about the impact of the work on a fundamental problem in spatial density estimation, and also notes that reviewers' concerns were sufficiently addressed in the response. The paper offers new loss based on optimal transport, and provides a theoretical contribution of generalization bounds, and SOTA results. Thus the AC recommends acceptance. Authors should update the paper according to the reviews and responses.